# Surface color and predictability determine contextual modulation of V1 firing and gamma oscillations

Alina Peter[1,5†]*, Cem Uran[1†], Johanna Klon-Lipok[1,2], Rasmus Roese[1], Sylvia van Stijn[1,2], William Barnes[1], Jarrod R Dowdall[1], Wolf Singer[1,3], Pascal Fries[1,4‡]*, Martin Vinck[1‡]*

[1]Ernst Strüngmann Institute (ESI) for Neuroscience in Cooperation with Max Planck Society, Frankfurt, Germany; [2]Max Planck Institute for Brain Research, Frankfurt, Germany; [3]Frankfurt Institute for Advanced Studies, Frankfurt, Germany; [4]Donders Institute for Brain, Cognition and Behaviour, Radboud University Nijmegen, Nijmegen, Netherlands; [5]International Max Planck Research School for Neural Circuits, Frankfurt, Germany

*For correspondence:
alina.peter@esi-frankfurt.de (AP);
pascal.fries@esi-frankfurt.de (PF);
martin.vinck@esi-frankfurt.de (MV)

[†]These authors contributed equally to this work
[‡]These authors also contributed equally to this work

**Competing interests:** The authors declare that no competing interests exist.

**Abstract** The integration of direct bottom-up inputs with contextual information is a core feature of neocortical circuits. In area V1, neurons may reduce their firing rates when their receptive field input can be predicted by spatial context. Gamma-synchronized (30–80 Hz) firing may provide a complementary signal to rates, reflecting stronger synchronization between neuronal populations receiving mutually predictable inputs. We show that large uniform surfaces, which have high spatial predictability, strongly suppressed firing yet induced prominent gamma synchronization in macaque V1, particularly when they were colored. Yet, chromatic mismatches between center and surround, breaking predictability, strongly reduced gamma synchronization while increasing firing rates. Differences between responses to different colors, including strong gamma-responses to red, arose from stimulus adaptation to a full-screen background, suggesting prominent differences in adaptation between M- and L-cone signaling pathways. Thus, synchrony signaled whether RF inputs were predicted from spatial context, while firing rates increased when stimuli were unpredicted from context.

DOI: https://doi.org/10.7554/eLife.42101.001

## Introduction

Visual processing relies on an integration of information over space, and an understanding of spatial relationships. This integration takes place in part in a feedforward manner through convergence of neurons with small receptive fields (RFs) onto neurons with larger RFs in higher areas (*Felleman and Van Essen, 1991*; *Serre et al., 2005*; *Lamme and Roelfsema, 2000*; *DiCarlo et al., 2012*). However, already in early visual cortex, neuronal responses to sensory inputs into the RFs are strongly modulated by the spatio-temporal context in which they are embedded. For instance, the firing rates of V1 neurons to stimuli in their classical RF (CRF), that is the region in space where stimuli have a strong driving effect, can be increased or decreased by stimuli presented in their surround (*Vinje and Gallant, 2000*; *Rao and Ballard, 1999*; *Angelucci et al., 2017*; *Gilbert, 1992*). In terms of anatomy, surround modulation represents a departure from feedforward processing. It is mediated by recurrent lateral and feedback connections, through which a given V1 neuron can be informed about a larger region of space than covered by its CRF (*Lund et al., 1993*; *Gilbert, 1992*; *Angelucci et al., 2017*). One view on surround modulation is that it merely represents a form of normalization, such that neuronal firing rates are essentially scaled by the amount of drive in the

surround (*Carandini and Heeger, 2011*). Additionally, it has been suggested that surround modulation may play an important role in various related functions like contour integration (*Liang et al., 2017*), perceptual filling-in (*Zweig et al., 2015*; *Land, 1959*; *Wachtler et al., 2003*), figure-ground separation (*Lamme, 1995*), computation of a saliency map (*Coen-Cagli et al., 2012*; *Li, 2002*), as well as efficient and predictive coding operations (*Rao and Ballard, 1999*; *Vinje and Gallant, 2000*).

Theories of efficient coding postulate that surround suppression of neuronal firing contributes to remove image redundancies across space from neuronal representations (*Schwartz and Simoncelli, 2001*; *Simoncelli and Olshausen, 2001*; *Coen-Cagli et al., 2012*; *Coen-Cagli et al., 2015*; *Rao and Ballard, 1999*; *Barlow, 2001*; *Vinje and Gallant, 2000*; *Zhu and Rozell, 2013*). Predictive coding theories hold that neuronal responses result from a comparison between predictions from the surround and the inputs into the CRF (*Friston, 2005*; *Rao and Ballard, 1999*; *Spratling, 2010*). Most studies on surround modulation have focused on the modulation of neuronal firing rates. However, if the modulation of neuronal firing rates arises from specific relationships between stimuli across space, then it might also have important consequences for temporal correlations among neuronal responses (*Singer and Gray, 1995*).

Recent work has extended the frameworks of efficient and predictive coding beyond firing rate modulations to include neuronal synchronization (*Bastos et al., 2012*; *Jadi and Sejnowski, 2014*; *Chalk et al., 2016*; *Vinck and Bosman, 2016*). Neuronal synchronization plays a functional role for the encoding and transmission of information, as well as for synaptic plasticity, and may therefore play an important role in contextual integration processes (*Buzsáki et al., 2012*; *Vinck et al., 2010a*; *Havenith et al., 2011*; *Salinas and Sejnowski, 2000*; *Simoncelli and Olshausen, 2001*; *Sejnowski and Paulsen, 2006*; *Fries, 2005*; *Fries et al., 2007*; *Fries, 2009*; *Börgers and Kopell, 2008*; *Kopell et al., 2000*; *Varela et al., 2001*; *Wang, 2010*; *Buzsáki, 2006*; *Cardin et al., 2009*; *Singer, 1999*; *Singer and Gray, 1995*; *Bressler et al., 1993*; *O'Keefe and Recce, 1993*; *Bernander et al., 1994*; *Abeles, 1982*; *Kempter et al., 1998*; *Azouz and Gray, 2000*; *Sohal et al., 2009*; *Ballard and Jehee, 2011*; *Ballard and Zhang, 2018*; *Akam and Kullmann, 2010*; *Akam and Kullmann, 2014*; *Palmigiano et al., 2017*). A distinguishing feature of V1 activity, induced by many stimulus conditions, is synchronization of neuronal activity in the gamma-frequency band ($\approx$ 30–80 Hz) (*Vinck and Bosman, 2016*; *Fries, 2009*; *Gray et al., 1989*; *Ray and Maunsell, 2010*; *Gieselmann and Thiele, 2008*; *Jia et al., 2013b*). A link between contextual modulation processes and V1 gamma-band synchronization is suggested by the finding that the amplitude of V1 gamma oscillations increases with stimulus size and therefore surround stimulation (*Gieselmann and Thiele, 2008*; *Ray and Maunsell, 2011*; *Jia et al., 2013b*; *Chalk et al., 2010*; *Perry et al., 2013*; *Jia et al., 2011*; *Gray et al., 1990*, for a detailed discussion, see *Vinck and Bosman, 2016*). There are different views on the way in which gamma oscillations might relate to predictive and efficient coding operations and therefore center-surround relationships (*Jadi and Sejnowski, 2014*; *Bastos et al., 2012*; *Arnal and Giraud, 2012*; *Vinck and Bosman, 2016*; *Chalk et al., 2016*; *Korndörfer et al., 2017*). *Bastos et al. (2012)* and *Arnal and Giraud (2012)* hypothesized that gamma-band synchronization subserves the encoding and transmission of prediction error signals in the feedforward direction, and that lower frequency bands carry feedback predictions from higher areas (*Bastos et al., 2012*; *Bastos et al., 2014*; *Arnal and Giraud, 2012*). Consistent with this hypothesis, bottom-up and top-down Granger-causal influences are strongest in the gamma and alpha/beta ($\approx$ 10–20 Hz) band, respectively (*Bastos et al., 2014*; *Richter et al., 2018*; *van Kerkoerle et al., 2014 Bressler et al., 2006*; *Bosman et al., 2012*; *Michalareas et al., 2016*). According to this hypothesis, a mismatch between center and surround stimuli should lead to an increase in both firing rates and gamma-band synchronization, conveying prediction error signals. In contrast, *Vinck and Bosman (2016)* recently hypothesized that (1) the amplitude of gamma oscillations in a given column reflects the extent to which classical RF inputs are predictable from the surround and (2) that gamma-band synchronization among columns with non-overlapping RFs reflects predictability among their visual inputs. This could in turn provide a mechanism for orchestrating interactions between distributed neuronal columns, and for integrating efficiently encoded signals in higher visual areas (*Vinck and Bosman, 2016*) (see Discussion). According to this hypothesis, redundancy between center and surround stimuli should lead to a decrease in firing rates (reflecting efficient coding) yet an increase in gamma-band synchronization. To distinguish between these conflicting views, precise manipulations of center-surround predictability are required.

As a starting point to test the interdependence between center-surround relationships, synchronization and firing rates, we considered the case of uniform surfaces and varied their size, center-surround relationships and color. The latter is an important feature for object recognition and visual search, and plays a role in social interactions and foraging (*Santos et al., 2001*; *Waitt et al., 2006*; *Gerald et al., 2007*; *Bichot et al., 2005*; *D'Zmura, 1991*; *Corso et al., 2016*). Despite the importance of the color domain in vision, colored stimuli have rarely been studied with respect to gamma synchronization (*Rols et al., 2001*; *Shirhatti and Ray, 2018*; *Brunet et al., 2015*). Furthermore, uniform surfaces are of particular interest because they contain highly redundant information across a relatively large image region. The predictability hypothesis (*Vinck and Bosman, 2016*) suggests that large, uniform surfaces should reliably induce gamma-band synchronization but be accompanied by low neuronal discharge rates. It further suggests that a modulation of center-surround predictability due to a chromatic mismatch (e.g. red center and green surround stimulus) should strongly reduce gamma-band synchronization but increase firing rates.

Uniform surfaces are characterized by hue, luminance and saturation and can be broadly divided into chromatic and achromatic (black and white) surfaces. Although uniform chromatic and achromatic surfaces both have a high degree of predictability at the (physical) image level, there are likely substantial differences in the way these surfaces are processed by area V1. Correspondingly, their predictability on the neuronal level likely differs. There are two different ways through which V1 surface representations may arise (*Zweig et al., 2015*). First, neurons with RFs at the uniform region of a surface stimulus (e.g. the center) may be directly activated. These may generate redundant (predictable) signals locally. Specifically for chromatic surfaces, single-opponent, hue-selective neurons (in LGN or V1) with RFs at the uniform surface region may directly encode color and luminance information. These are neurons with L+/M-, M+/L-, or blue (S) and yellow (L and M) color opponencies (*Shapley and Hawken, 2011*; *Livingstone and Hubel, 1984*). Second, surface information may be encoded by neurons with RFs at the edge of the surface, and then propagate toward neurons with RFs at the uniform region of the surface (*Zweig et al., 2015*; *Land, 1959*; *Wachtler et al., 2003*). The relative contributions of these two mechanisms (local vs. edge-derived) remain largely unknown and likely differ between chromatic and achromatic surfaces (*Zweig et al., 2015*; *Zurawel et al., 2014*). *Zweig et al. (2015)* have addressed this question by making voltage-sensitive dye recordings of V1 populations and comparing the responses at the surface's center to the edge (*Zweig et al., 2015*). Activity patterns for achromatic surfaces were consistent with an edge-derived 'fill-in' process (*Zweig et al., 2015*). However, this was not observed for chromatic surfaces (*Zweig et al., 2015*), which could be due to the availability of another, surface rather than edge-based, information source provided by single-opponent cells. Because contextual interactions likely have a different nature for achromatic than chromatic stimuli, we asked whether there are differences in the contextual modulation of firing activity and gamma-band synchronization between these two classes of stimuli.

In this study, we investigated the contextual modulation of V1 firing activity and gamma-band synchronization using chromatic and achromatic surfaces of different sizes, and a center-surround mismatch paradigm. Additionally, we examined differences among color hues, adaptation over time, and the influence of the full-screen background (FSB) on which surfaces were displayed.

## Results

We recorded multi-unit (MU) activity and local field potentials (LFP) from the primary visual cortex (area V1) in two macaque monkeys, while they performed a fixation task. These recordings were made using a 64-channel chronic microelectrode array in monkey H and a 32-channel semichronic microelectrode array in monkey A (see Materials and methods). Classical receptive fields (RFs, referring to classical RFs unless otherwise mentioned) of the MU activity were estimated using moving bar stimuli (see Materials and methods; monkey H: median RF eccentricity 6.2 deg, range 5.2–7.1 deg, median RF diameter 0.48 deg, range 0.26–1.88 deg; monkey A: median eccentricity 5.4 deg, range 3.2–8.5 deg, median RF diameter 0.91 deg, range 0.46–2.3 deg). Compared to a surface stimulus of 6 deg diameter, receptive fields had a median proportional diameter of 0.08 (0.04–0.31, monkey H) or 0.15 (0.08–0.38, monkey A). We first studied LFP and MU responses to the presentation of stationary surface stimuli, namely large uniform disks covering the cluster formed by all RFs (6 deg diameter, flashed on and then maintained on screen; *Figure 1A–B*; *Dataset 1*, see Materials and methods). The stimuli did not overlap with the fixation spot. Note that the stimuli were much larger

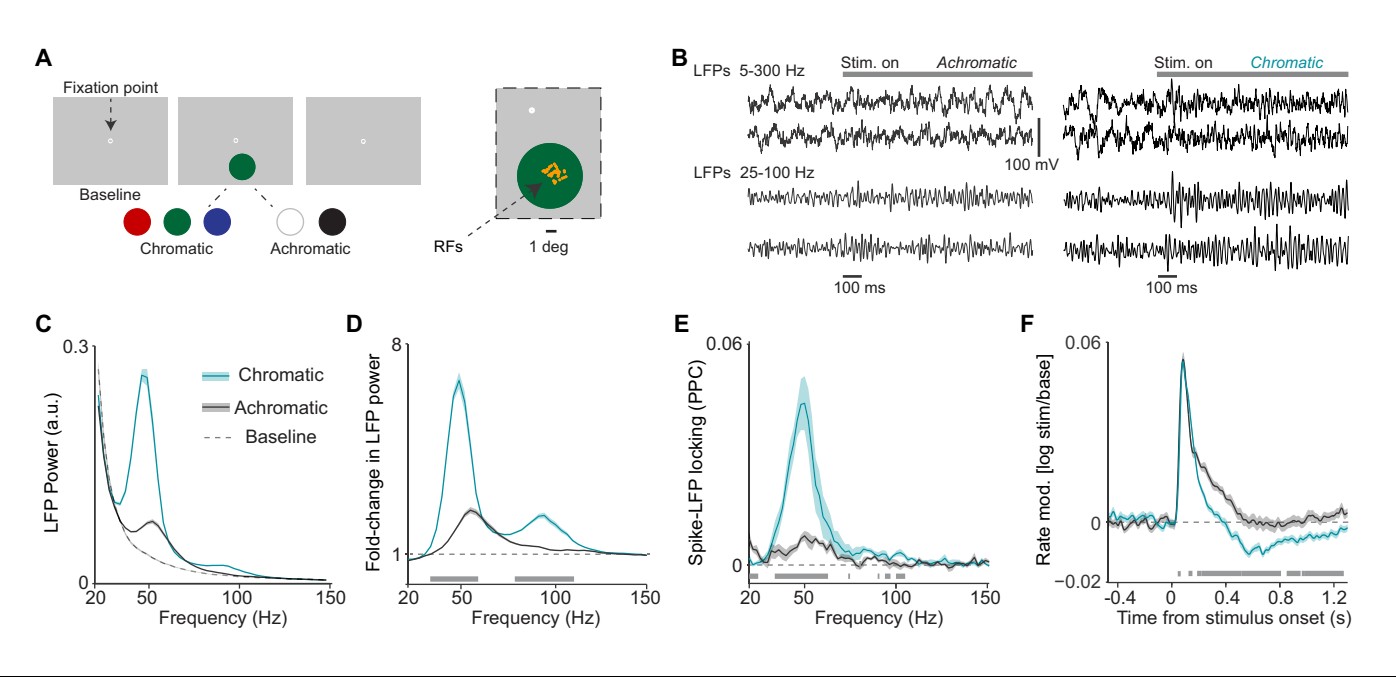

**Figure 1.** Analysis of LFP and multi-unit activity in response to large, uniform surfaces. (**A**) Illustration of experimental paradigm with large, 6 deg diameter surfaces (*Dataset 1*, see Materials and methods; n = 5 sessions, 60±0 trials per session for chromatic and 40±0 trials per session for achromatic conditions. Trials numbers were equated by random subselection for statistics.). Trials were self-initiated by fixating on the fixation spot (enlarged for visibility), followed by a baseline period of 0.5–0.8 s with a gray background screen. Surfaces were either chromatic (red, blue, or green) or achromatic (black or white) and presented for 3.3 s, the first 1.3 s of which are analyzed here. Right panel shows the RF locations of analyzed sites in one session. (**B**) Representative trials of LFP signals for achromatic and chromatic conditions (having gamma power close to the median of the respective condition). (**C**) Average LFP power spectra for chromatic (turqoise), achromatic (black) and baseline (gray) conditions. LFP power was estimated using Discrete Fourier Transform of non-overlapping epochs of 500 ms, with multi-tapering spectral estimation (±5 Hz). LFP spectra for all three conditions were normalized to the summed power (>20 Hz) for the baseline (gray) condition (see Materials and methods). (**D**) Average change in LFP power, expressed as fold-change, relative to baseline. (**E**) Average MU-LFP locking, which was estimated using the pairwise phase consistency (PPC, see Materials and methods). (**F**) Modulation of firing rate relative to baseline, expressed as $\log_{10}(stim/base)$. (**D–F**) Shadings indicate standard errors of the means obtained with bootstrapping (see Materials and methods). Gray bars at bottom of figure indicate significant differences between chromatic and achromatic stimuli, obtained from permutation testing with multiple comparison correction across all frequencies and time points (see Materials and methods).

DOI: https://doi.org/10.7554/eLife.42101.002

The following figure supplements are available for figure 1:

**Figure supplement 1.** Illustration of fitting procedure.

DOI: https://doi.org/10.7554/eLife.42101.003

**Figure supplement 2.** Analysis of LFP and multi-unit activity in response to large, uniform surfaces.

DOI: https://doi.org/10.7554/eLife.42101.004

**Figure supplement 3.** Control analysis for microsaccades.

DOI: https://doi.org/10.7554/eLife.42101.005

than the RFs of the multi-units, such that they covered a large portion of the multi-units' respective surround regions.

Initially, we analyzed differences between chromatic and achromatic surface stimuli (*Figures 1* and *2*; see Materials and methods). We then considered the specific differences among responses to distinct color hues and achromatic stimuli (Figures 4–7). The LFP power spectra had similar frequency profiles in the two monkeys (i.e. the peaks were well aligned; *Figure 1—figure supplement 2*), and the MUs showed similar temporal profiles (*Figure 1—figure supplement 2*). Therefore, we pooled the data from the two animals. Note that statistical parameters are largely described in the figure captions.

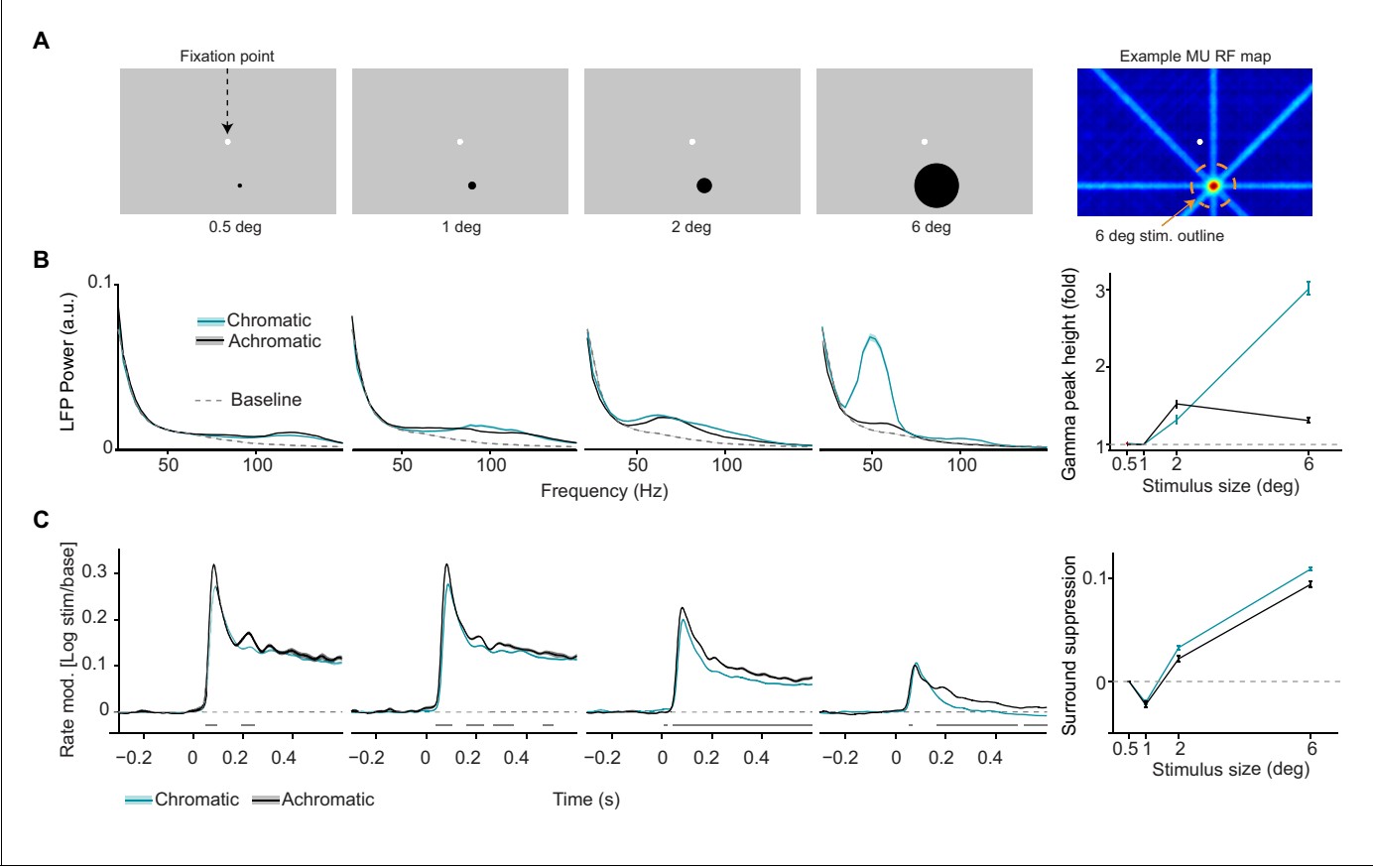

**Figure 2.** Dependence of LFP power spectra and MU firing activity on surface size. (**A**) Illustration of experimental paradigm (*Dataset 2*, see Materials and methods; n = 9 sessions, 59.75±0.09/25.64±0.12 trials chromatic vs achromatic trials per condition in each session). Uniform surfaces of four different sizes were presented on a gray background screen. Fixation spot is enlarged for visibility. Right: Receptive field estimated with bar stimuli for a representative target channel, with the outline (orange dashed line) of the largest size stimulus (see Materials and methods). Note that the activation outside the RF is due to the use of large bar stimuli sweeping over the monitor. (**B**) LFP power spectra for different sizes and chromatic/achromatic conditions. LFP power spectrum estimated and normalized as in *Figure 1C*, but now using 300 ms epochs. Right panel shows the gamma-band amplitude as a function of size, estimated using a polynomial fitting procedure between 30 and 80 Hz (see Materials and methods). The difference between 6 and 0.5 deg stimuli was significantly larger for chromatic than achromatic condition (P<0.05, bootstrap test, see Materials and methods). (**C**) Modulation of firing rate relative to baseline, expressed as $\log_{10}(stim/base)$, for different sizes and chromatic/achromatic conditions. Right panel shows surround suppression, which was defined as the difference in firing rate modulation between the 0.5 degree size and the other sizes.
DOI: https://doi.org/10.7554/eLife.42101.006

The following figure supplement is available for figure 2:

**Figure supplement 1.** Analysis of LFP and multi-unit activity in response to stimuli of varying size.
DOI: https://doi.org/10.7554/eLife.42101.007

## Characteristics of firing activity and LFP signals in response to uniform surface stimuli

We examined the effect of uniform surface stimuli on LFP power spectra. The presentation of large, chromatic surface stimuli (equiluminant red, green and blue; see Materials and methods) induced prominent, narrow-band gamma oscillations in LFP power spectra (*Figure 1C–D*). These oscillations were clearly visible in the LFP traces (*Figure 1B*). In comparison, gamma-band oscillations were significantly weaker for achromatic surface stimuli (black or white, maximal contrast to background; *Figure 1C–D*). This finding was highly consistent across sites. For each site and chromatic/achromatic condition, we determined the peak power change in the gamma-frequency range (30–80 Hz) using a polynomial fit (see neuronal and Materials and methods). Gamma peak power changes were stronger for chromatic than achromatic surface stimuli at 97.8% (45 out of 46) of LFP recording sites

(*Figure 1—figure supplement 2*). Note that in the Section 'Controls for luminance-contrast and cone-contrast', we will describe the results of control experiments in which the chromatic and achromatic stimuli are matched in terms of luminance-contrast and DKL cone-contrast (see Materials and methods) to the FSB. We find that the difference in gamma-band oscillations between chromatic and achromatic stimuli is not explained by either luminance or cone-contrast to the FSB. We also removed 100 ms data epochs after each microsaccade and found that the LFP results on gamma oscillations were qualitatively unchanged (*Figure 1—figure supplement 3*).

To test whether V1 spiking activity was synchronized with the induced LFP gamma oscillations, we computed spike-field phase locking spectra (Pairwise Phase Consistency, *Vinck et al., 2010b*) between MU and LFP activity obtained from nearby but separate sites (*Figure 1E*; see Materials and methods). Spike-field phase-locking spectra for chromatic surface stimuli showed a prominent peak in the gamma-frequency band consistent with the gamma peak in the LFP power spectrum (*Figure 1D*), whereas phase-locking was significantly weaker for achromatic surface stimuli (*Figure 1E*). The main point of this analysis is that gamma-synchronization for chromatic stimuli is not merely observed at the level of synaptic currents within V1, but also at the level of V1 output spikes.

Next, using the same stimulus paradigm, we examined the way in which the presentation of uniform surface stimuli affected MU firing activity. The presentation of chromatic and achromatic surface stimuli induced short-latency onset transients of similar magnitude (*Figure 1F*). However, we observed a stronger decrease in MU firing activity over time during continuous stimulus presentation for chromatic than achromatic surface stimuli, starting around 200 ms after the stimulus onset (*Figure 1F*). Strikingly, for chromatic surface stimuli, MU firing activity fell below baseline levels (*Figure 1F*). Note that in Figure 4D, we show that the decrease in MU firing below baseline only occurred for a subset of colors. The reduction in MU firing rates (0.3–1.3 s period) for chromatic as compared to achromatic surface stimuli was observed for 92% of recording sites. Control analyses in which data epochs after microsaccades were removed indicate that the late decrease in MU firing was not due to microsaccades (*Figure 1—figure supplement 3*).

These findings demonstrate that large, uniform, chromatic surface stimuli induce low firing activity yet highly gamma-synchronous V1 responses, whereas achromatic surface stimuli induce much weaker gamma-band synchronization but relatively more vigorous firing activity (for further interpretation of this finding, see Discussion).

## Dependence of firing activity and LFP signals on stimulus size

The results shown in *Figure 1* are consistent with the predictability hypothesis (*Vinck and Bosman, 2016*) outlined in the Introduction. Yet, they do not demonstrate directly that the enhancement in gamma-band synchronization observed for large uniform colored surfaces is due to contextual surround modulation, because we did not manipulate the surround input. Furthermore, it remains unclear whether the observed differences between chromatic and achromatic surfaces can be explained by a difference in contextual surround modulation or other factors like stimulus drive. To address these questions directly, we used a paradigm that varied the stimulus size across trials (*Figure 2A*; see Materials and methods). We selected one site (or a few nearby sites with RF centers within 0.5 deg of the target site) per session and centered the stimulus on the multi-unit's RF, which was previously mapped with moving bars. In each trial, a stimulus of a particular size (0.5, 1, 2 or 6 deg diameter) was presented for 600 ms (*Figure 2A–B*).

We first examined how the characteristics of LFP power spectra depended on stimulus size. Analysis of LFP power spectra revealed a strong dependence of gamma power on stimulus size for chromatic stimuli, and by comparison a much weaker dependence for achromatic stimuli (*Figure 2B*). To quantify this size dependence, we determined the gamma peak power between 30 and 80 Hz (as described for *Figure 1*). For chromatic stimuli, increases in stimulus size resulted in increases in induced gamma peak power as soon as the stimulus also covered the surround (i.e. from 2 deg onwards, *Figure 2B*). By contrast, for achromatic stimuli, a gamma peak in the 30–80 Hz band emerged from 2 deg stimulus size onwards and showed no further increase with stimulus size. Given the relatively broad increase in >100 Hz LFP power seen in *Figure 2B*, we also determined gamma peak power and peak frequency in a wider range (30–150 Hz). This analysis revealed LFP power peaks >100 Hz for the sizes below 2 deg, and again the strong size dependence for chromatic

compared to achromatic stimuli (*Figure 2—figure supplement 1A*; also see *Figure 2—figure supplement 1B* for an analysis per animal).

We further investigated the way in whichMU firing was modulated by surround stimulation. We observed that for both achromatic and chromatic stimuli, MU firing activity was highest for 0.5–1 deg stimulus sizes (*Figure 2C*). This was consistent with the estimates obtained from RF mapping and the fact that we centered the presented stimuli on the MUs' estimated RFs. For small stimuli (0.5–1 deg), only the initial transient in MU firing activity showed a difference between chromatic and achromatic conditions, with slightly higher firing activity for achromatic than chromatic stimuli (*Figure 2C*). In contrast, the presentation of a 2 or 6 deg stimulus, increasingly covering the surround, induced strong suppression of MU firing activity as compared to the 0.5 deg stimulus (*Figure 2C*, rightmost panel). This surround suppression was stronger for chromatic than achromatic stimuli (*Figure 2C*).

Furthermore, we analyzed responses during a later period in the trial, when the small stimulus had been presented for 600 ms, and a large (6 deg) surface stimulus of the same color was added for another 600 ms period (*Figure 2—figure supplement 1C*). We found that this addition of the surround stimulus alone induced a rapid suppression of MU firing activity, which was significantly more pronounced for chromatic than achromatic stimuli (*Figure 2—figure supplement 1C*).

These findings suggest that the relatively strong decrease in firing over time observed for large, chromatic surfaces (*Figure 1*) is at least partially explained by surround suppression. They furthermore indicate that for the small RF stimuli, there are no substantial differences on average between chromatic and achromatic surfaces in terms of MU and LFP responses. Yet, we find a prominent difference in the way chromatic and achromatic stimuli are affected by surround stimulation.

## Modulation of firing activity and LFP signals by center-surround predictability

A potential explanation for the results shown in *Figures 1–2* may be the center-surround predictability hypothesis outlined in the Introduction (*Vinck and Bosman, 2016*). Yet, the employed paradigm used stimuli of different sizes, which may have recruited different neuronal circuits and may also have changed stimulus salience. We therefore used an additional stimulus paradigm in which surround influences were modified, while stimulus size was held constant. Specifically, we created three sets of equally sized stimuli. In one set, the surround was fully predictive of the RF stimulation, because it used a uniform surface (called 'uniform' stimulus). In the second set (called 'blob mismatch'), the surround was not predictive of the RF stimulation, because the surround stimulus and the 1 deg RF stimulus had different colors (which were physically equiluminant). In the third set (called 'annulus mismatch'), the surround had the same color as the RF stimulation, but the two were separated by an annulus ring of a different, physically equiluminant color. This annulus ring had 0.25 deg thickness and an inner diameter of 1 deg.

We found that compared to the uniform surfaces, stimuli with a chromatic (blob or annulus) mismatch had higher MU firing activity (*Figure 3C*). This held true both for the initial transient period and the subsequent sustained response period (*Figure 3C*). At the same time, we observed a marked decrease in the amplitude of LFP gamma oscillations for the chromatic mismatch compared to the uniform surface stimuli (*Figure 3B–C*). This result was consistent across animals (*Figure 3—figure supplement 1A*).

We further investigated whether this pattern of changes was specific to the sites having RFs near the center stimulus. To this end, we examined sites with RFs on the outer uniform regions of the stimulus (with RF centers 1.5–2 deg from the stimulus center; *Figure 3D*). For these sites the MU firing responses did not differ significantly between conditions during the initial transient period (*Figure 3D*). During the later sustained response phase, however, MU firing activity was reduced for the chromatic mismatch stimuli compared to the uniform surface stimulus (*Figure 3D*). Note that whenever the RF center covered a large uniform surface region, either in the uniform stimulus condition or when it covered the surround region of the mismatch stimuli, sustained firing levels were below baseline. This confirmed the respective finding reported in *Figure 1*.

These results suggest that a mismatch between stimuli at the RF center and the surround can dramatically change the surround influence on responses to the center. We wondered whether the surround influence on gamma oscillations originates from the uniform surface region or rather from the edge of the surface. To this end, we analyzed sessions in which we compared two sets of trials: First,

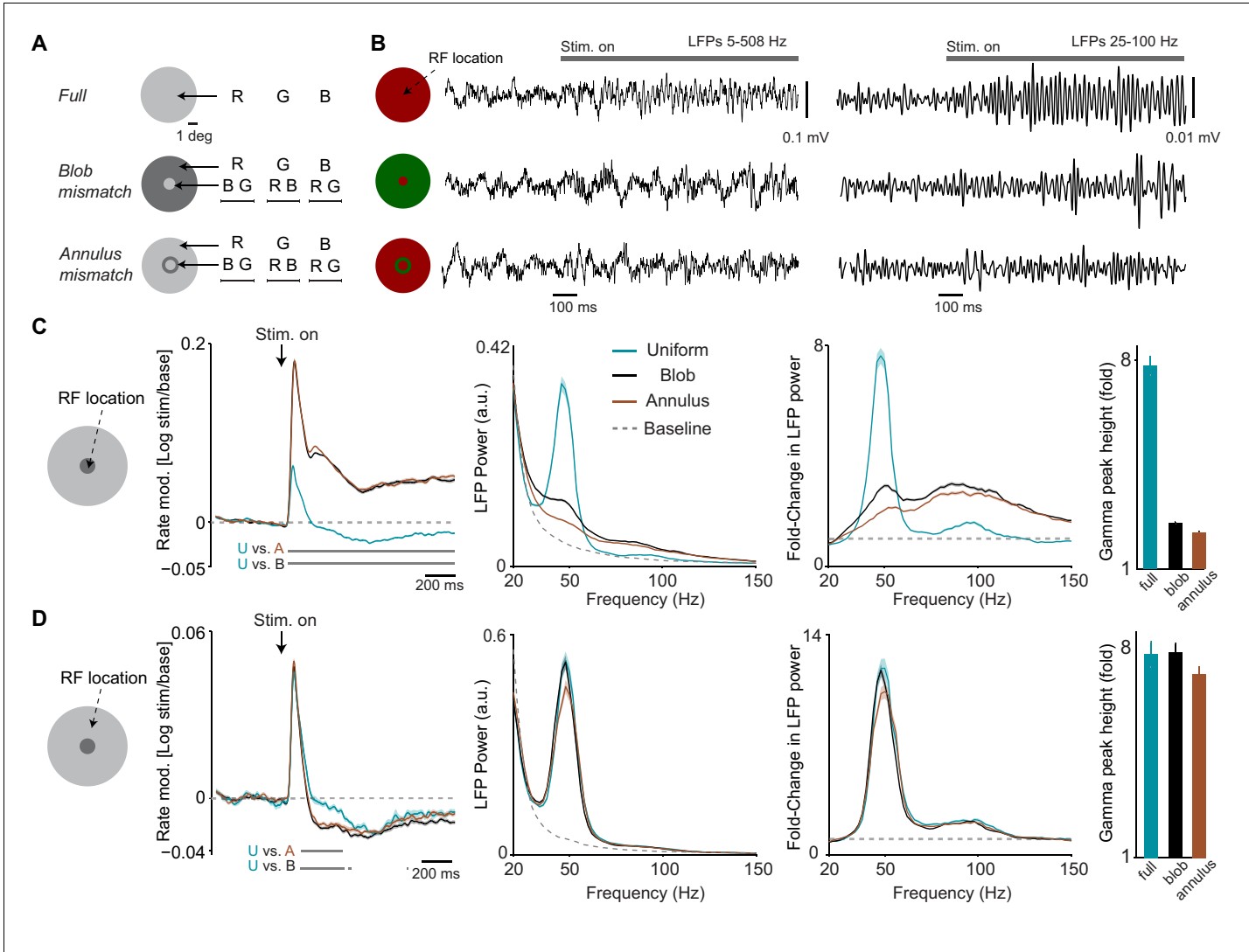

**Figure 3.** Dependence of LFP power spectra and firing rates on spatial predictability. (**A**) Illustration of paradigm (*Dataset 3*, see Materials and methods; n = 8 sessions, 16.26±0.19 trials in each of the 15 conditions per session). We grouped stimuli intro three types. In the 'Uniform surface' group of conditions, stimuli of 6 deg diameter were presented with either a red, blue or green hue (R B G, equiluminant). In the second 'blob mismatch' group, the central 1 deg of the stimulus had a different (equiluminant) color than the rest of the stimulus. In the third 'annulus mismatch' group, we presented an annulus ring (of 0.25 deg) of another color on top of the uniform surface (at equiluminant intensity) around the inner one degree from the stimulus center. All combinations of hues and stimulus types were presented, yielding a total of 15 individual conditions. (**B**) Representative LFP traces (having gamma power close to the median of all trials for the respective condition) for the three stimulus types. (**C**) Analysis for target channels with RFs at the center of the stimulus. Shown from left to right are: (1) The change in MU firing activity relative to baseline expressed as $\log_{10}(stim/base)$. (2) LFP power spectra for the three stimulus conditions and the baseline. LFP power spectrum estimated and normalized as in *Figure 1C*. (3) The change in LFP power relative to baseline, expressed as a fold-change. (4) The gamma-band amplitude, estimated using a polynomial fitting procedure (see Materials and methods). Gamma-band amplitude was significantly higher for uniform surface than blob and annulus conditions (p<0.05, bootstrap test, see Materials and methods). (**D**) Same as (**C**), but now for target channels with RFs between 1.5 and 2 deg from the stimulus center, that is close to the central region of the larger, uniform region of the stimulus. Gamma-band amplitude did not significantly differ between conditions (bootstrap test, all P>0.08 ).

DOI: https://doi.org/10.7554/eLife.42101.008

The following figure supplement is available for figure 3:

**Figure supplement 1.** Additional analyses and experiments performed in relationship to *Figure 3* in the main text.

DOI: https://doi.org/10.7554/eLife.42101.009

trials with a full surface stimulus centered on a site's RF ('RF-on-center' condition; *Figure 3—figure supplement 1B*). Second, trials with a full surface stimulus positioned such that its edge fell into the RF center, that is with the surface shifted by 3 deg horizontally ('RF-on-edge' condition; *Figure 3—figure supplement 1B*). We found that the amplitude of gamma oscillations was significantly higher at the center ('RF-on-center') than at the edge of the surface stimulus ('RF-on-edge'), whereas the opposite was observed for MU firing activity (*Figure 3—figure supplement 1B*). In one session (monkey H), we also showed disk stimuli that had their edge blurred with a Gaussian (2.5 deg size, 1 deg standard deviation). There were clear gamma-responses also in this case (*Figure 3—figure supplement 1C*).

Together, these results indicate that for colored surfaces the amplitude of gamma-band oscillations is commensurate with the 'chromatic' predictability among visual inputs in space, and that gamma-band oscillations are not a mere consequence of input drive to a larger cortical region. Furthermore, these results suggest that gamma strength can be dissociated from stimulus salience, because the chromatic mismatch condition provided a highly salient stimulus in the RF, but resulted in weaker gamma.

## Differences in firing activity and LFP signals between color hues

The results above show prominent differences between chromatic and achromatic surfaces in terms of gamma-band synchronization. The respective analyses pooled different chromatic conditions (equiluminant red, green and blue) together. However, there may exist further differences in gamma-band synchronization within the chromatic conditions, that is between different hues. To investigate this we used two types of stimulus sets, which were presented in separate sessions. In the first stimulus set (*Figure 4A*), we presented each surface color at its maximum possible luminance level (given the limits of the employed monitor), and sampled from the entire spectrum of hues available with the monitor (see Materials and methods). In the second stimulus set, we presented surface stimuli with different color hues at equated luminance levels (*Figure 4B–C*).

Using the first stimulus set, we found that gamma-band LFP oscillations were reliably induced across the entire spectrum of hues (*Figure 4A*, *Figure 4—figure supplement 1*, see also *Supplementary file 1* for all luminance and CIE values). In addition, we found that gamma-band synchronization was reliably induced by surfaces with 'extra-spectral' colors, that is colors resulting from a mixture of blue and red primaries (*Figure 4A*), as well as brownish hues. We further replicated our finding that gamma oscillations were relatively weak for both black and white surface stimuli as compared to all colored surfaces (*Figure 4A*). In one monkey (A), we found that gamma-band activity was stronger for black than for white stimuli, consistent with previous results showing stronger firing rate responses to black than white stimuli (*Xing et al., 2010*; *Yeh et al., 2009*). However, a trend in the opposite direction was observed for monkey H (*Figure 4—figure supplement 2*).

For the first stimulus set (*Figure 4A*), the different colors were presented at their maximum possible luminance levels, which might confound the effects of hue and luminance. We therefore used a second stimulus set in which we presented surface stimuli with different color hues at three levels of equal physical luminance, that is different color values (*Figure 4B–C*). For all three hues, gamma amplitudes were greater for the highest compared to the lowest luminance condition (p<0.05, bootstrap test, see Materials and methods; *Figure 4B–C*). The dependence of gamma amplitude on stimulus luminance was greater for green than for red or blue surface stimuli (difference between high versus low, *Figure 4B–C*). Gamma oscillations had a higher amplitude for red than for blue or green surface stimuli across all three luminance conditions (*Figure 4B–C*), whereas gamma amplitude was higher for blue than green surface stimuli for low and intermediate luminance conditions (*Figure 4(B–C)*). Another difference between the hues was that the gamma peak had a significantly lower frequency for green compared to red or blue surface stimuli (p<0.05, bootstrap test; *Figure 4B–C* and *Figure 4—figure supplement 2*). The results of these analyses were consistent across both monkeys (*Figure 4—figure supplement 2*).

Given the relationships between MU firing activity and LFP gamma-band oscillations shown in *Figures 1–3*, we asked how these differences in LFP gamma oscillations were related to changes in firing activity. During the initial transient, MU firing activity was higher for red and blue rather than for green surface stimuli, with slightly stronger responses for red than blue surface stimuli (*Figure 4D*). Yet, we found that the post-transient decrease in MU firing activity over time was particularly pronounced for red and particularly weak for green stimuli (*Figure 4D*). In agreement with

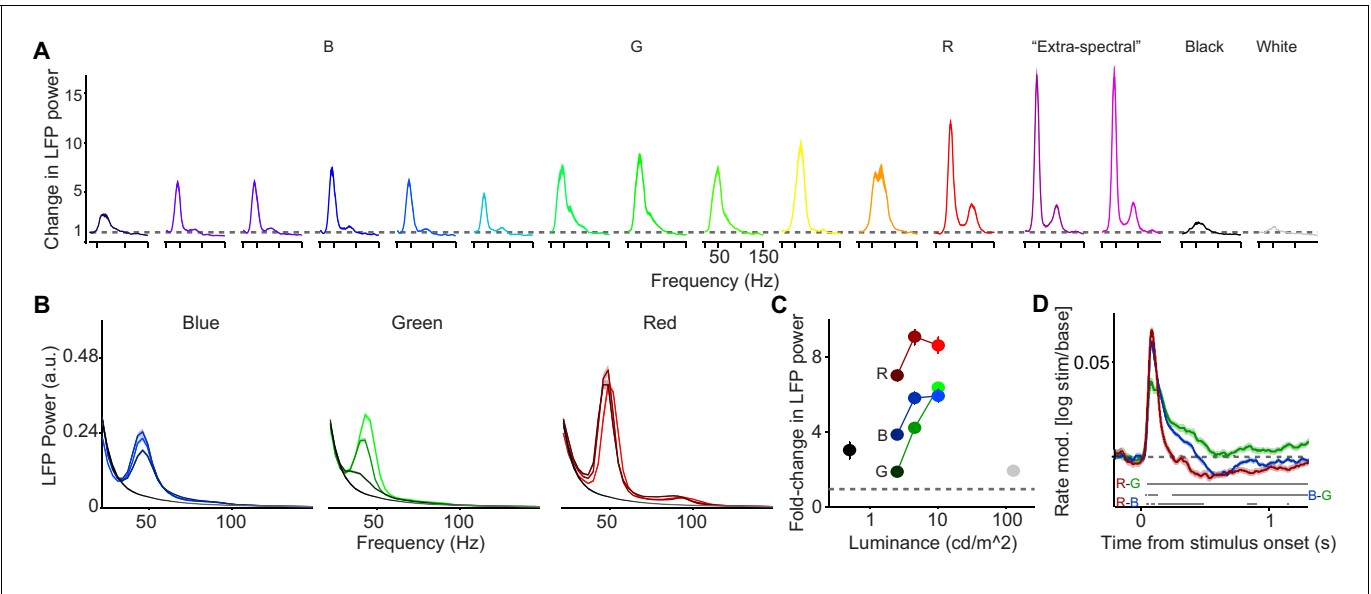

**Figure 4.** Dependence of LFP power spectra and MU firing activity on surface hue and luminance. (**A**) We presented uniform surfaces of 6 deg at maximum possible luminance levels, sampling from the available spectrum of wavelengths (*Dataset 4*, see Materials and methods; n = 5 sessions, 15.24±0.1 trials per condition in each session ). In addition, we presented black and white surfaces. Shown is the change in LFP power relative to the baseline (gray screen), expressed as a fold-change. (**B**) Three hues (red, green and blue) were presented at three different luminance levels (approximately 2.5, 5 and 10 cd/m$^2$, *Dataset 1*, see Materials and methods; n = 5 sessions, 19.8±0.45 trials per condition in each session). Shown are LFP power spectra. LFP power spectrum estimated and normalized as in *Figure 1C*. (**C**) Average change in LFP power, expressed as a fold-change based on the fitting procedure, relative to the baseline (gray screen). The dependence of gamma amplitude on stimulus luminance was greater for G than for R or B (difference between high versus low: p<0.05, bootstrap test). Gamma oscillations amplitude R>B or G across all three luminance conditions, and B>G surface stimuli for low and intermediate luminance conditions (P<0.05, bootstrap test). (**D**) Modulation of firing rate relative to baseline, expressed as $\log_{10}$(stim/base). Horizontal bars at bottom of panel represent significant differences between stimuli at p<0.05 (permutation test, multiple comparison corrected for time bins). (**B–D**) Color hues were adjusted for better discriminability, panel A of *Figure 4—figure supplement 2* shows actual hues.

DOI: https://doi.org/10.7554/eLife.42101.010

The following figure supplements are available for figure 4:

**Figure supplement 1.** Dependence of gamma LFP power on the surface hue: additional analyses in relation to main *Figure 4A*.
DOI: https://doi.org/10.7554/eLife.42101.011

**Figure supplement 2.** Dependence of gamma LFP power on the surface hue: further analyses in relation to main *Figure 4B*.
DOI: https://doi.org/10.7554/eLife.42101.012

**Figure supplement 3.** Dependence of gamma LFP power on the surface hue: control experiment for luminance-contrast.
DOI: https://doi.org/10.7554/eLife.42101.013

**Figure supplement 4.** Gamma-band power for stimuli defined on equiluminant DKL planes, control experiment related to *Figure 4* of main text.
DOI: https://doi.org/10.7554/eLife.42101.014

the data shown in *Figure 1*, we observed that MU firing activity fell below baseline levels for red and blue surface stimuli (*Figure 4D*).

Together, these results indicate that surfaces of all color hues tend to induce gamma-band oscillations with a higher amplitude compared to achromatic surfaces, and that the amplitude of gamma oscillations is relatively high for red surfaces.

## Controls for luminance-contrast and cone-contrast

In the analyses above, we observed a strong difference in gamma-band power between chromatic and achromatic surfaces. We performed several control analyses and experiments to investigate whether this observed difference was explained by differences in DKL cone contrast or luminance contrast between chromatic and achromatic surfaces. A linear regression of gamma peak height against absolute Michelson luminance contrast (luminance stimulus - luminance baseline / (luminance stimulus +luminance baseline)) across the surface stimuli shown in *Figure 4B* showed no significant

relationship (r = −0.44, p=0.16, F-test, *Figure 4—figure supplement 2*; note that the relationship, if any, was negative). In an additional control experiment, we directly matched the luminance (and thereby luminance-contrast) of the achromatic and chromatic stimuli across five brightness values, including the FSB brightness and two steps of positive and negative contrast. We found that achromatic gamma-responses were much weaker than chromatic gamma-responses regardless of overall luminance level, also under these matched conditions (*Figure 4—figure supplement 3A*). We additionally used this experiment as a control for the effect of pupil size (see Materials and methods) on gamma-band amplitudes (*Figure 4—figure supplement 3B*). Note that gamma responses for achromatic stimuli were weak regardless of the degree of pupil change.

In another experiment, which is part of the data shown in Figure 6, we matched cone-contrasts between chromatic and achromatic stimuli. Specifically, we compared gamma-responses to a colored surface on an achromatic FSB with gamma responses to a corresponding achromatic surface on a chromatic FSB of the same respective color (e.g. red on a gray background versus gray on a red background). These comparisons keep the changes in cone-activation relative to the background the same. Note that this does not mean that the cone-contrasts are matched in the DKL space, because this space contains an additional normalization step, which incorporates the extent to which the FSB itself activates the different cones. Nevertheless, although only the non-normalized changes in cone-contrasts are matched, it can be seenthatfor example the white stimuli have very strong DKL cone-contrast to the chromatic FSBs along the L-M and S-(L + M) axes (*Figure 6—figure supplement 1*). This cone-contrast for white surfaces on chromatic backgrounds exceeds that of chromatic stimuli on the white background (*Figure 6—figure supplement 1*). Our analyses reveal that for each tested color (except for red on a black surface), gamma was much stronger for chromatic than achromatic surfaces of matched cone contrast (*Figure 6F*). Together, these data indicate that the difference in gamma-band response between chromatic and achromatic surfaces was not due to luminance- or DKL cone-contrast relative to the FSB.

In the previous section, colored stimuli were either presented at maximum brightness or presented at the same physical luminance. We performed additional experiments in which colored surfaces were matched in terms of DKL space coordinates in units of Weber cone contrast (see Materials and methods, *Figure 4—figure supplement 4*). These coordinates were the L-M (red-green opponency), S-(L + M) (blue-yellow opponency) and L + M (luminance) cone-contrasts relative to the gray FSB. In the first experiment (*Figure 4—figure supplement 4A*), we selected three luminance steps (L + M cone-contrast was −0.25, 0, or +0.25). For each luminance step, we then took an equal step in the positive and negative L-M direction. This step was taken as the maximum possible step for which the magnitudes were equal in both directions. Similarly, we took a step of equal magnitude in the positive or negative S-(L + M) direction. In the second experiment (*Figure 4—figure supplement 4B*), we sampled from eight different angles in the DKL plane at an equiluminant level to the gray background. Note that this yields stimuli that are highly desaturated as compared to the stimuli shown in *Figure 4*.

The data from these control experiments show that gamma-band responses were stronger for reddish than greenish hues. This is consistent with the results shown in *Figure 4A* (see *Figure 4—figure supplement 1D*). Achromatic stimuli did not induce detectable gamma-band response peaks, neither in the positive or negative luminance-contrast (L + M) step, consistent with the findings of *Figures 1–2* and *4*. Furthermore, these data suggest that increasing S-(L + M) cone-contrast (blue-yellow opponency) independent (or in absence of) L-M cone-contrast also boosts gamma oscillations (*Figure 4—figure supplement 4B*). These results further support the notion that gamma-band oscillations for uniform surfaces are mediated by color-opponency signals. They further indicate that hue (i.e. the angle in the DKL-plane) itself is a determinant of gamma-band oscillation strength and that the dependence on hue is not explained by the magnitude of cone-contrasts. This is also consistent with the finding that the magnitudes of the DKL cone-contrasts for the chromatic surface stimuli shown in *Figure 4A* are not significantly correlated with gamma-band power (*Figure 4—figure supplement 1C*). Note that this lack of a correlation might be due to the use of stimuli with largely very high brightness and cone contrasts, such that the effects of hue differences dominate.

## Temporal evolution of gamma-band responses

The observed differences in gamma oscillations between the color hues (*Figure 4*) might reflect a static and context-independent property of visual cortex to respond differently to distinct hues. Yet,

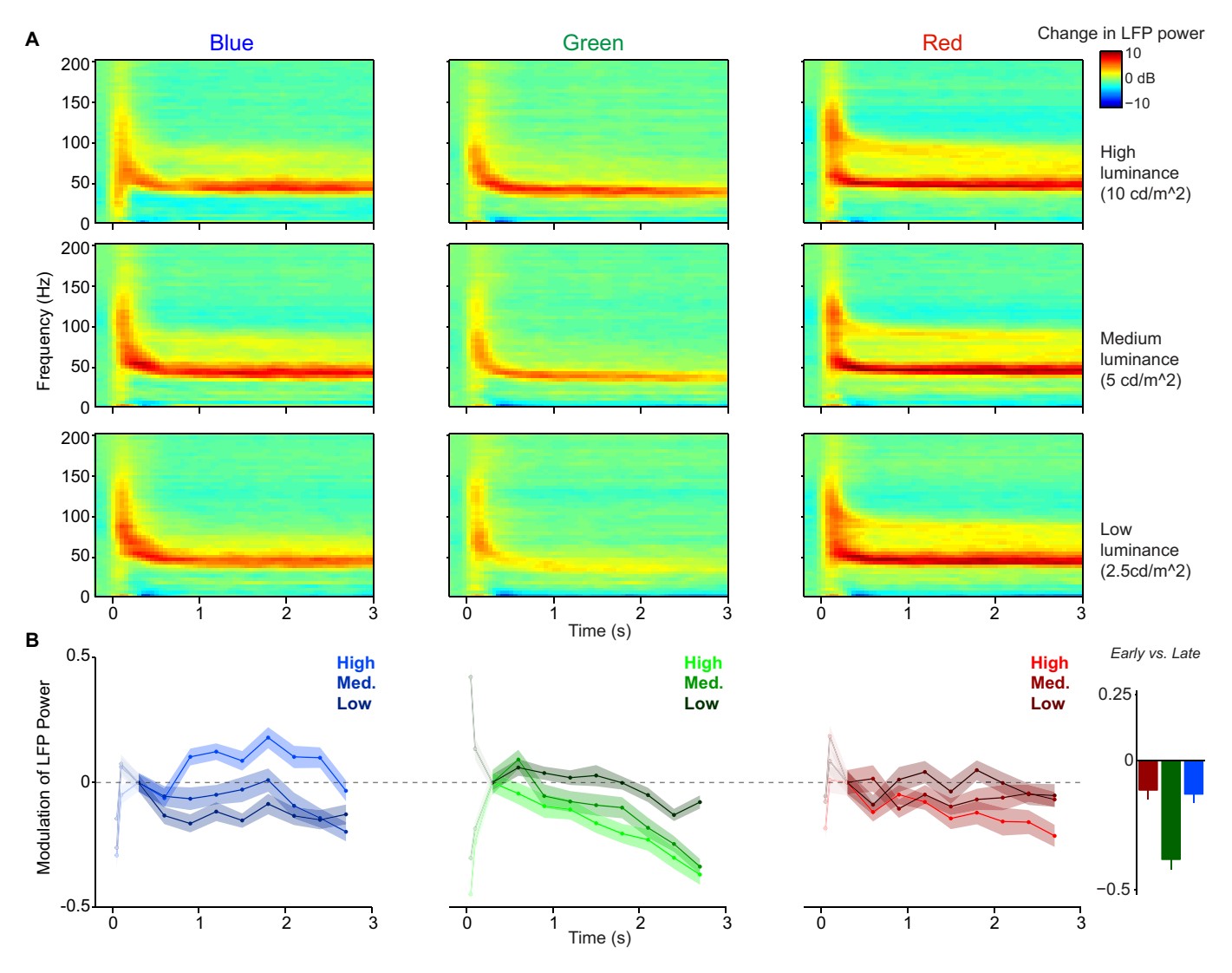

**Figure 5.** Within-trial temporal dynamics of LFP power spectra during viewing of uniform surfaces. Time-frequency representations of log-transformed change in LFP power relative to baseline (dB), using *Dataset 1* (n = 5 sessions, 19.8±0.45 trials per condition in each session). Shown are the three (equiluminant) different color hues at three different luminance levels. (B) Change in LFP power relative to 0.3–0.6 s period (third time point), separate for different color hues and luminance levels, shown as fold-change modulation index (see Materials and methods). The first two points correspond to 0.05–0.35 and 0.1–0.4 s and are whitened out because these points are strongly influenced by initial firing transient and associated bleed-in of spiking activity at high frequencies. Right panel shows the modulation of LFP power in early (0.3–0.6 s) versus late (2.7–3.0 s) period, averaged over three luminance levels. The decrease in gamma peak amplitude over time was significantly larger for green than blue and red surfaces (peak amplitude estimated as described in Materials and methods, main effect across luminance levels: p<0.05, bootstrap test), and did not differ between blue and red conditions. This also held true for the modulation of LFP power for the highest luminance condition only (p<0.05, bootstrap test).

DOI: https://doi.org/10.7554/eLife.42101.015

the continuous presentation of a uniform surface stimulus for the duration of an entire trial likely induces substantial adaptation at many levels of the nervous system. We thus wondered whether different hues might adapt at different rates. To address this, we examined the temporal evolution of LFP power spectra over a longer time period, that is up to 3 s after stimulus onset. Time-frequency representations showed that qualitative differences between hues and between luminance levels tended to be relatively stable over time (*Figure 5A*). However, we found that the amplitude of gamma-oscillations decreased more rapidly over time for green than for blue or red surface stimuli (*Figure 5A–B*). This result held also for both animals individually (monkey H/A red change over time

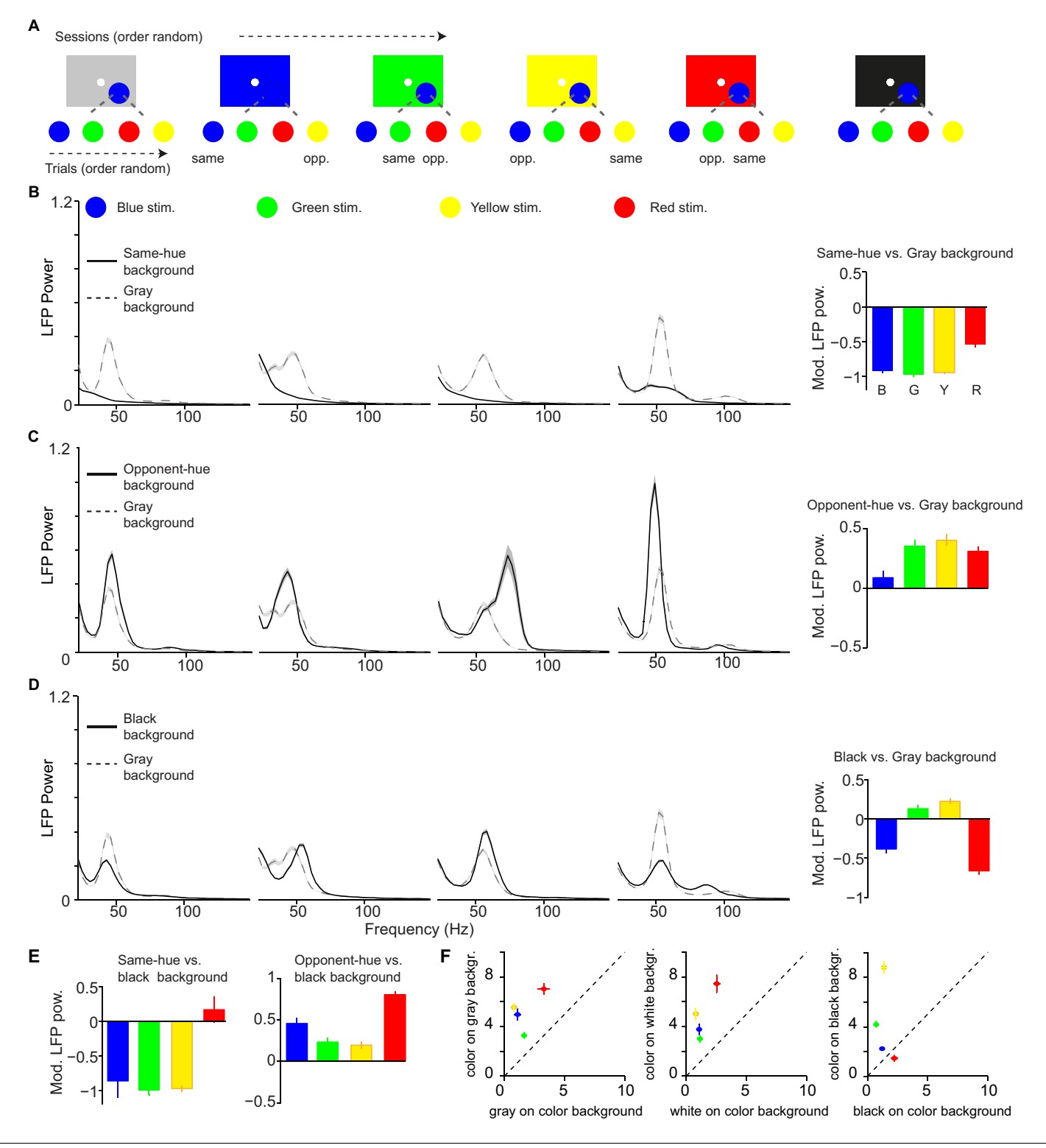

**Figure 6.** Dependence of LFP power spectra on background stimulus. (**A**) Illustration of paradigm (*Dataset 5*, see Materials and methods; n = 25 sessions, 18.64±0.11 trials per condition and session). In a given session, a fixed background stimulus was used, and a set of chromatic and achromatic surfaces (6 or 8 deg) were presented in separate trials (see Materials and methods). (**B**) Average LFP power spectra for the different color conditions during gray background versus same-hue background sessions. We show analyses for blue, green, yellow, and red surfaces, presented at maximum possible luminance. Right: modulation index of LFP gamma-amplitudes (see Materials and methods). Main effect: p<0.05, bootstrap test. R versus G, B

*Figure 6 continued on next page*

*Figure 6 continued*

or Y, and G versus B: p<0.05, bootstrap test. (**C**) As B) for comparison of opponent color background and black background condition Main effect: p<0.05. B versus R, G or Y, p<0.05. (**D**) black background versus gray background. Main effect not significant. All color differences significant (p<0.05, bootstrap test). (**E**) Modulation of gamma-band amplitude for same-hue vs black background condition (left), as well as opponent-hue vs black background condition (right). Left: p<0.05: R versus G, B or Y; B versus G or Y Right: p<0.05: all combinations except G versus Y. (**F**) Comparison of gamma-band responses between chromatic surfaces shown on achromatic background and achromatic surfaces shown on the same respective chromatic background (using the data shown in *Figure 6—figure supplement 2*).

DOI: https://doi.org/10.7554/eLife.42101.016

The following figure supplements are available for figure 6:

**Figure supplement 1.** DKL-space representation for *Figure 6* (main text) and *Figure 6—figure supplement 2*.

DOI: https://doi.org/10.7554/eLife.42101.017

**Figure supplement 2.** Dependence of gamma LFP power on the combination surface hue and background stimulus.

DOI: https://doi.org/10.7554/eLife.42101.018

**Figure supplement 3.** LFP power spectra for the post-stimulus period for the four chromatic background hues (3.5–3.8 s, excluding the initial transient response after stimulus offset at 3.3 s).

DOI: https://doi.org/10.7554/eLife.42101.019

$-0.07\pm0.05/-0.12\pm0.04$, green $-0.51\pm0.06/-0.36\pm0.04$, blue $-0.23\pm0.05/-0.10\pm0.04$). The main effect of decrease with time, as well as stronger decreases for green compared to both red and blue, were signicant in both animals individually (all p < 0.05 corrected for multiple comparisons).

This suggests that there may be differences in the time course and strength of adaptation between color hues, specifically stronger adaptation for green surface stimuli.

## Dependence on full-screen background hue

One potential source of adaptation, other than the surface stimulus of a given trial, is the color composition of the continuously presented background. In the experiments described above, all surface stimuli were displayed on a gray FSB. Gamma-band responses to achromatic and chromatic surface stimuli may have been affected by the use of this gray FSB, given that the FSB itself may induce adaptation at many levels of the nervous system. We therefore asked how gamma-band responses to surface stimuli depend on the color of the FSB. To answer this question, we performed experiments in which we used different FSBs in separate, adjacent sessions (gray, white, black, blue, green, yellow and red) (*Figure 6A* and *Figure 6—figure supplement 1*; see Materials and methods). The FSB was continuously presented during the entire session, that is remained on both during the pre-stimulus period, post-stimulus period and the period during which the surface stimuli were displayed (*Figure 6A*). In *Figure 6*, we analyze LFP responses to the presentation of chromatic surface stimuli of different hues, which were presented at the maximum possible luminance level (see *Figure 6—figure supplement 2* for equiluminant red, green and blue as well as achromatic surface stimuli).

We first examined how the responses to surface stimuli with specific hues (e.g. green) were altered by using an FSB with the same hue (e.g. green), comparing them to the sessions with a gray FSB (*Figure 6B*). Surprisingly, we found that when the FSB had the same hue as the surface stimulus, there was a nearly complete abolishment of gamma oscillations for blue, green and yellow stimuli (*Figure 6B* and *Figure 6—figure supplement 2*). This was observed both when the surface stimulus had the same luminance as the background (*Figure 6*) and when the surface stimulus had a lower luminance (*Figure 6—figure supplement 2*). Interestingly, red stimuli could still induce detectable gamma oscillations when presented on a red FSB, although the gamma amplitude was strongly reduced compared to the gray FSB condition (*Figure 6B* and *Figure 6—figure supplement 2*).

The reduction in gamma-band oscillations for the same-hue FSB condition may have been an effect of stimulus size, because the background is effectively a very large surface. Alternatively, it may have been an effect of stimulus history. To investigate these possibilities, we analyzed the post-stimulus period immediately following the offset of a gray surface stimulus that was displayed on a colored FSB. We found that the reappearance of the FSB after the offset of the gray surface induced prominent gamma-band oscillations (*Figure 6—figure supplement 3*). This indicates that the decrease in gamma-band oscillations with the same hue FSB condition was not due to the large size of the background color stimulation, but that it was due to the continuous presence of the same-hue background. This is also consistent with a previous report showing strong gamma with full-screen

color stimuli that change color across trials (*Shirhatti and Ray, 2018*), and with the positive relation between gamma and stimulus size shown in *Figure 2C*.

Next, we considered interactions between distinct hues. We wondered whether gamma oscillations can not only be reduced by same-hue FSBs, but also enhanced by FSB hues that are different from the stimulus hue, in particular when FSB and stimulus assume opponent colors. The organization of color vision around color-opponency axes, namely the red-green and the blue-yellow axes, is a key principle found both at the neurophysiological and psychophysical level (*Wachtler et al., 2003*; *Livingstone and Hubel, 1984*; *Solomon and Lennie, 2007*; *Tailby et al., 2008a*). These color opponencies are thought to result from the computation of differences among signals deriving from L and M cones (red-green), and S cones versus L and M cones (blue-yellow). We found that for all surface hues, gamma oscillations were amplified when stimuli and FSBs were of opponent color hues (*Figure 6C*). This suggests that gamma oscillations are dependent on opponency signals along the red-green and the blue-yellow axes (*Figure 6C*). Given the strong dependence of gamma-band oscillations on the FSB, we asked whether the use of a gray FSB may have induced differences in gamma-band amplitude among distinct hues. To examine this possibility, we used a black FSB, which should induce minimal adaptation for all cones. Quite surprisingly, the difference among red, green and blue hues that we had observed with a gray FSB could not be replicated when we presented the stimuli on a black FSB (*Figure 6D*). Compared to the gray FSB condition, gamma-band amplitudes were significantly lower for red and blue surface stimuli and significantly higher for yellow and green surface stimuli (*Figure 6D*). As a consequence, for the black FSB condition, gamma-band power was no longer highest in response to red stimuli, but showed a different dependence on hue (*Figure 6D*; *Figure 6—figure supplement 2*). Specifically, gamma-band power was higher for green and yellow than red and blue surface stimuli (*Figure 6D*). The resulting pattern could not be explained by luminance contrast differences, because contrast increased for all hues on the black compared to the gray FSB, whereas gamma increased for some hues and decreased for others (*Figure 6D*).

In *Figure 6B–C*, we compared stimulus responses in the same-hue and opponent-hue background conditions with stimulus responses in the gray background condition. However, because of the evidence that the gray FSB may not have affected all stimulus hues equally, we also directly compared the same-hue and opponent-hue FSB conditions with the black FSB condition. This analysis revealed a marked difference between red and the other hues (*Figure 6E*). First, when an FSB of the same hue as the stimulus was compared to a black FSB, gamma was almost abolished for blue, green and yellow, but not for red stimuli. Second, when an FSB of the opponent hue was compared to a black FSB, gamma was enhanced for all colors, but particularly strongly for red (*Figure 6E*). The full matrix of gamma responses for different FSB conditions in *Figure 6—figure supplement 2* shows that for all non-red chromatic FSBs, gamma oscillations were strongly amplified for red surface stimuli.

We also analyzed the gamma-responses to achromatic stimuli on colored backgrounds, and asked in particular whether responses of achromatic stimuli on colored surfaces were as strong as responses of colored stimuli on achromatic surfaces (*Figure 6F* ). Achromatic responses on colored backgrounds were substantially weaker than the reverse (see Section 'Controls for luminance-contrast and cone-contrast'). These data demonstrate that gamma oscillations depend strongly on the FSB, in a way that follows the color-opponency axes. Furthermore, a commonly used 'default' of the display, namely gray, introduces adaptation effects that are color-specific.

## A quantitive model relating hue dependence of gamma-band oscillations to adaptation

To explain how gamma-band responses to surface stimuli depend on the FSB, we constructed a quantitative model by estimating the degree to which each FSB differentially adapts the S-, M- and L-cone pathways. Note that this model is agnostic to the neuronal locus at which adaptation of the distinct cone pathways occurs, for example it might occur in the retina, LGN or visual cortex. We hypothesize that gamma-band oscillations for colored surface stimuli are mediated by the activation of single color-opponent cells in a large spatial region by the same color input. The combination of bottom-up drive at each point of the surface and strong surround modulation may then lead to gamma oscillations. The result that gamma oscillations are particularly strong in the opponent-hue background condition (*Figure 6*) further suggests that when this circuit is more strongly activated

(leading to stronger input drive as well as stronger surround modulation), gamma oscillations increase.

Following this reasoning, we further hypothesized that the dependence of gamma-band oscillations on the FSB can be explained by adaptation of specific cone pathways (see Discussion for further argumentation). As an example, a green FSB should lead to stronger adaptation of the M-cone compared to the L-cone pathway. This should increase the degree to which single-opponent cells with L+/M- color-opponencies are activated by red surface stimuli, which may in turn increase the amplitude of gamma-band oscillations.

To capture this intuition in a quantitative manner, we constructed a model in which we aimed to predict the difference in gamma-band amplitudes between red and green surface stimuli (for blue and yellow surface stimuli see further below). The variable to be predicted was the red-green gamma ratio, defined as $\gamma_{ratio} = \log_{10}(\gamma_{red}/\gamma_{green})$, where $\gamma_{red}$ and $\gamma_{green}$ are the respective gamma-amplitudes for red and green surface stimuli. This $\gamma_{ratio}$ was computed separately for all the different FSBs. We estimated the degree to which each FSB adapts the M- and L-cones, using the known response curves of the three cones as a function of wavelength from macaque monkeys (*Hárosi, 1987*) (see Materials and methods, *Figure 7A*). We then measured the physical wavelength spectrum for each FSB as realized on our monitor. We multiplied the FSB spectra of the different color primaries with the response functions of each cone and summed over wavelengths. This yielded for each FSB stimulus two parameter values, $M_{adapt}$ and $L_{adapt}$. We then fitted a multiple regression model predicting $\gamma_{ratio}$ from $M_{adapt}$ and $L_{adapt}$ plus a constant regression intercept (*Figure 7B*). For this model, we used response data for both green and red surfaces presented at maximum possible luminance, as well asequiluminant red and green surfaces, across the different FSBs.

The regression analysis reveals that $\gamma_{ratio}$ can be highly accurately predictedbythe way in which each FSB adapts the L and the M cones (*Figure 7B*; $R^2 = 0.91$, $p < 0.05$, F-Test). The regression coefficients for $M_{adapt}$ and $L_{adapt}$ were positive and negative, respectively. This indicates that adaptation of the M-cone increases $\gamma_{ratio}$, whereas adaptation of the L-cone decreases $\gamma_{ratio}$ (*Figure 7B*). We also found that the$\gamma_{ratio}$ could not be significantly predicted when using $S_{adapt}$ and $L_{adapt}$ as predictors (p=0.23), consistent with the idea that the neuronal mechanisms underlying the red-green opponency are dependent on the M versus L cone contrast. The regression intercept of the model (on the $\gamma_{ratio}$ axis) was not significantly different from zero. This indicates that green and red tend to generate gamma oscillations of similar amplitude when the FSB does not adapt the cones, consistent with the findings shown for the black FSB (*Figure 6*). Strikingly, we found that the $M_{adapt}$ coefficient had an absolute magnitude approximately twice as large as the $L_{adapt}$ coefficient (*Figure 7B*). This suggests that uniform surfaces tend to adapt the M-cone pathway more strongly than the L-cone pathway, or that adaptation of the M-cone pathway has a stronger effect on gamma-band oscillations than adaptation of the L-cone pathway.

The model further explains some non-trivial findings that would have been unexpected if FSBs had affected the M- and L-cone pathway in a similar way: We observed that the yellow FSB strongly amplified $\gamma_{ratio}$ (*Figure 7B*). Given its wavelength spectrum, the yellow FSB is expected to adapt the L-cones more strongly than the M-cones, which would predict a reduced $\gamma_{ratio}$,that is red responses being weaker than green responses. By contrast, we found that $\gamma_{ratio}$ was enhanced. The models explain this by the fact that the stronger L-cone than M-cone activation by the yellow background is more than compensated by the much greater $M_{adapt}$ than $L_{adapt}$ coefficient. Similarly, the $\gamma_{ratio}$ increased for a gray compared to a black FSBs, even though gray FSBs should in principle adapt the M- and L-cone pathways to a similar degree (*Figure 6*). This was again compensated by the much greater $M_{adap}$ than $L_{adap}$ coefficient.

We performed a similar analysis for the yellow-blue (L + M S) opponency axis, aiming to predict the gamma ratio of blue over yellow (*Figure 7C*). We first fitted a model with the S, L and the M cone parameters, and $\gamma_{ratio}$ was now defined as $\gamma_{ratio} = \log_{10}(\gamma_{blue}/\gamma_{yellow})$. This regression model explained a large degree of variance ($R^2 = 0.99$), with almost equal magnitude of S (negative, $-1.23$) and L (positive, 1.24) coefficients, but a much smaller and non-significant coefficient for the M cone ($-0.29$). The finding that the model fit included a highly positive L-cone coefficient and non-significant (and negative) M-cone coefficient seems *prima facie* to contradict the canonical idea that the perceptual blue-yellow opponency axis is mediated by an S versus (L + M) opponency. However,

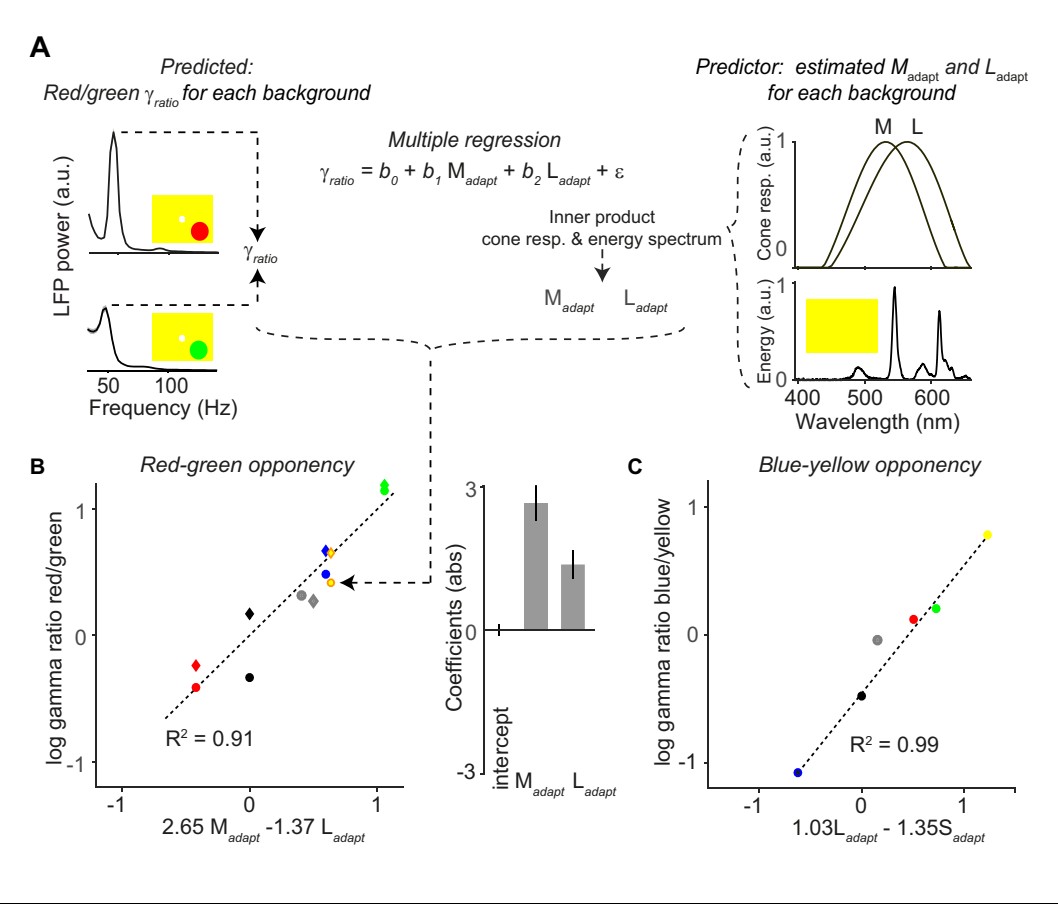

**Figure 7.** Quantitative model for dependence of gamma-band amplitude on background stimulus. (**A**) With a multiple regression model, we predicted the ratio of gamma amplitude for red over green surface stimuli. Shown are the LFP gamma power spectra for red (top) and green (bottom) surface stimuli, with a yellow FSB. Normalized cone responses for macaque monkeys are shown on the right, constructed by fitting polynomials to bleaching difference spectra data (*Hárosi, 1987*). For each full-screen background (FSB), we estimates the extent to which it adapted the three cones ($M_{adapt}$, $L_{adapt}$ and $S_{adapt}$; see Materials and methods), by convolving the spectral energy of the FSB with the normalized cone response curves. For illustration purposes, we only show the M- and L-cone response curve. (**B**) The hues of the data points correspond to the FSB, and the shape indicates whether the red and green surfaces were presented at maximum possible luminance intensity (circle), or whether they were presented at equiluminant intensities (diamond). As predictor variables we used $M_{adapt}$ and $L_{adapt}$ for the different FSBs. The dependent variable was the red/green gamma-ratio, $\gamma_{ratio}$. The coefficients indicates the relative influence of the two adaptation parameters on the $\gamma_{ratio}$, and the sign indicates whether adaptation of a given cone is increasing or decreasing the red/green gamma-ratio. (**C**) Similar to (**B**), but now for blue-yellow. In this case we used $S_{adapt}$ and $L_{adapt}$ as prediction parameters. A model using $M_{adapt}$ in addition did not yield a significant coefficient for $M_{adapt}$.

DOI: https://doi.org/10.7554/eLife.42101.020

The following figure supplement is available for figure 7:

**Figure supplement 1.** Analysis of *Figure 7* in main text performed separately for the two monkeys.
DOI: https://doi.org/10.7554/eLife.42101.021

neurophysiological data has shown that the main opponency for LGN cells on the yellow-blue axis is the L versus S cone (*Tailby et al., 2008b*). We further simplified our model using two predictive parameters, equating the blue axis to the S cone and the yellow axis to the L cone (*Figure 7C*). Again, we found a highly predictive relationship with negative weight for the S cone and a positive weight for the L cone (*Figure 7C*), with only a small and non-significant difference in the magnitude of the adaptation coefficients. The results were qualitatively highly similar between animals and individually significant (*Figure 7—figure supplement 1*). These findings indicate that gamma oscillations

are mediated not only by opponency signals along the red-green axis, but also along the blue-yellow axes, consistent with the data shown in *Figure 4—figure supplement 4*.

## Discussion

### Summary

We investigated the way in which V1 responses to chromatic and achromatic surfaces are modulated by spatial and temporal context. We report the following main findings:

1. Compared to achromatic surfaces, chromatic surfaces induced strong synchronization of neuronal activity in the gamma-frequency band (*Figures 1* and *2*). This finding held true for color hues across the entire wavelength spectrum (*Figure 4*).
2. Whereas chromatic and achromatic surfaces induced an initial MU firing transient of similar magnitude (*Figures 1* and *2*), we found relatively weaker firing responses to chromatic surfaces in the sustained stimulation period, which was evident from a stronger decrease in firing over time related to an increase in surround suppression (*Figures 1* and *2*).
3. Compared to uniform chromatic surfaces, composite stimuli with a chromatic mismatch between the stimulus covering the RF center and the surrounding surface evoked high firing activity, yet induced a very prominent reduction in the amplitude of gamma band oscillations (*Figure 3*). This supports the hypothesis of *Vinck and Bosman (2016)* outlined in the Introduction, namely that gamma-band synchronization reflects the degree to which inputs into the CRF can be predicted from the surround.
4. Stimulus-induced gamma-band oscillations were also strongly modulated by the larger spatio-temporal context: We found that their amplitude depended strongly on the FSB on which the surfaces were displayed (*Figures 4–7*). We concluded that the dependence of gamma-band synchronization on the FSB was explained by two key factors: First, color opponency along one of the two main color-opponency axes (red versus green and yellow versus blue). Second, a comparatively stronger adapting influence of FSBs on the M-cone pathway, which leads to comparatively stronger gamma oscillations for red surfaces for many (but not all) FSBs (*Figures 6–7*). This is consistent with our finding that the decrease in gamma-band amplitude within a trial is particularly strong for green surface stimuli (*Figure 5*).

### Differences between chromatic and achromatic surfaces

We asked whether there are differences in V1 gamma-band synchronization and firing responses between chromatic and achromatic stimuli. A previous voltage-sensitive dye imaging study has shown differences in V1 responses to chromatic compared to achromatic surfaces (*Zweig et al., 2015*). For achromatic surfaces, V1 response characteristics were indicative of a fill-in process, in which surface information emanates from the surface's edge. This was not the case for chromatic surfaces (*Zweig et al., 2015*), which suggests that the V1 representation of surface color may largely depend on single-opponent LGN inputs to V1 and responses of V1 neurons with RFs within the uniform surface (*Zweig et al., 2015*; *Livingstone and Hubel, 1984*; *Shapley and Hawken, 2011*). Indeed, the data shown in *Figures 6–7* suggests that V1 gamma oscillations are mediated by color-opponency signals. It has been shown that a subset of neurons with chromatic opponencies in LGN and V1 exhibit elevated firing as compared to baseline for large chromatic surfaces, and carry information about the surface's hue (*Ts'o and Gilbert, 1988*). These cells can be subdivided into Type I, Type II and modified Type II responses. These color-selective neurons would not be active for a large achromatic stimulus (*Ts'o and Gilbert, 1988*), indicating that there may be stronger LGN and/or layer four cortical drive for chromatic than achromatic stimuli. However, a subset of V1 neurons also fire to temporal luminance changes for black and white surfaces (*Xing et al., 2010*), and neurons might also be activated for achromatic surfaces by fill-in processes from the surround (*Zweig et al., 2015*). This means that chromatic and achromatic stimuli may differ in input drive, and perhaps other aspects of processing (*Zweig et al., 2015*). Several human imaging studies have shown stronger fMRI signals in response to chromatic stimuli (for review, see *Schluppeck and Engel, 2002*; *Shapley and Hawken, 2011*). Because the fMRI signal correlates not only with spiking activity, but also with gamma responses (*Logothetis and Wandell, 2004*; *Ekstrom, 2010*; *Maier et al., 2008*; *Thomsen et al., 2004*; *Viswanathan and Freeman, 2007*; *Nir et al., 2007*; *Scheeringa et al., 2016*,

for visual gamma in particular *Bartolo et al., 2011*, *Niessing et al., 2005*), it remains unclear how these fMRI findings are related to the present findings.

In the present study, we observed that chromatic surfaces exhibited much stronger gamma-band synchronization yet more suppressed firing than achromatic surfaces. This finding is consistent with the idea that V1 representations of surface color depend on the direct activation of neurons with RFs in the uniform region of the surface. It further supports the hypothesis that gamma-band synchronization arises from the predictability of visual inputs across space (*Vinck and Bosman, 2016*). Together, these data suggest that one could think of color, like stimulus orientation, as a 'feature', with predictive value for stimulus features in color space. If the features at the center stimulus are correctly predicted by the surround (context), this results in strong gamma-band synchronization. On the other hand, predictability of stimulus luminance by itself in the absence of color information, as in the case of a large uniform white stimulus, may not be processed as a stimulus feature. Luminance by itself, in the absence of color information, may not be sufficient for inducing strong gamma-band synchronization. In contrast, prominent gamma-band synchronization can be generated in response to achromatic stimuli when they have structural features (orientation, frequency, phase) that are highly predictable over space, for example bars and gratings (*Gieselmann and Thiele, 2008*; *Gray et al., 1989*; *Chalk et al., 2010*; *Gail, 2000*; *Singer, 2018*).

Overall, our data suggests that both sufficient drive and spatial predictability are the necessary ingredients for the generation of V1 gamma. There are several cases in our manuscript where differences in V1 gamma are dissociated from differences in firing rate. For example, in *Figure 3* we show a strong increase in firing rates for the annulus and blob condition as compared to the uniform surface; however, gamma oscillations are markedly decreased. In *Figure 3—figure supplement 1B*, we show that gamma-band oscillations are stronger at the center than the edge of a chromatic surface stimulus; however, firing rates are much stronger at the edge of the chromatic surface stimulus. In V1, most cells with chromatic opponencies are found in the superficial layers, to which our recordings are biased and in which gamma is thought to be generated (see Discussion section 'Mechanisms of gamma-band synchronization'). Most of these V1 neurons are strongly driven by the presence of chromatic edges and have band-pass rather than low-pass spatial characteristics, and the majority of neurons has been classified as double-opponent rather than single-opponent (*Shapley and Hawken, 2011*; *Johnson et al., 2008*; *Friedman et al., 2003*; *Shapley and Hawken, 2002*). This is consistent with our finding that firing rates are relatively high at the edge of the chromatic surface stimulus and for the blob and annulus mismatch conditions (*Figure 3*, *Figure 3—figure supplement 1B*). As opposed to these cases where high firing rates are accompanied by weak gamma, we also present cases where stimuli have high spatial predictability, but where gamma is weak or absent likely due to low drive (*Figure 1* achromatic stimuli, *Figure 6* chromatic stimuli after prolonged adaptation).

An important question is how to functionally relate the dependence of gamma on both predictability and drive. Notably, even though chromatic and achromatic (uniform) surfaces both have high spatial predictability at the *image level*, they may differ in the degree of predictability at the level of inputs at the *neuronal level*. Before information about an image reaches the cortex, it is processed through various stages. Noise could accumulate through these processing steps, for example due to synaptic release noise, ion channel noise and background synaptic activity. We can sketch two extreme cases as a function of the signal-to-noise ratio, which we call the efficient coding (1) and the inference regime (2) (*de Lange et al., 2018*; *Rao and Ballard, 1999*): (1) In the efficient coding regime where the signal-to-noise ratio is high and the image has predictable relationships over space, redundant information should be removed by a subtraction of predictions. The removal of redundant information should lead to a sparse code (*Rao and Ballard, 1999*), accompanied by gamma-synchronization. This may be mediated by GABAergic, inhibitory mechanisms (*Jadi and Sejnowski, 2014*; *Vinck et al., 2013b*) (see the Discussion section on mechanisms below). (2) If the sensory input is relatively weak, as in case of a stimulus with low luminance-contrast, the signal-to-noise ratio is expected to be low. In such a case, there is less redundancy between center and surround inputs, that is less predictability. The principle of Bayesian inference tells us that in these kind of conditions, the surround may effectively be used to infer representations at the RF location (*Rao and Ballard, 1999*; *de Lange et al., 2018*). In other words, the representation in this case is biased toward the contextual prediction, and is essentially a weighting of the input with the contextual prediction (*Rao and Ballard, 1999*; *de Lange et al., 2018*). The weighting of contextual surround

information with the input may in this case rely on an increase in local firing driven by excitatory surround influences, and a concurrent decrease in the recruitment of GABAergic interneurons by surround inputs (*Rao and Ballard, 1999*; *Jadi and Sejnowski, 2014*). Consistent with these ideas, previous work has shown surround facilitation and an expansion of the classical receptive field under low luminance-contrast condition (*Kapadia et al., 1995*; *Kapadia et al., 1999*; *vonder Heydt et al., 1984*; *Grosof et al., 1993*), and perception is biased towards expectation under low signal-to-noise ratio conditions (for a review see *de Lange et al., 2018*). Furthermore, it has been shown that gamma-band synchronization for grating stimuli increases with luminance-contrast (*Ray and Maunsell, 2010*; *Henrie and Shapley, 2005*; *Roberts et al., 2013*; *Hadjipapas et al., 2015*). For achromatic stimuli, responses at the surface's center derived from local signals may be weak. Responses should therefore be influenced by excitatory contextual influences from neurons with receptive fields on the surface's edge (*Zweig et al., 2015*). In contrast, the surface representation for chromatic stimuli may largely depend on the direct activation of single-opponent cells with RFs on the uniform region of the surface. In this case, there is high redundancy across space and the efficient coding process dominates, which is accompanied by gamma-synchronization. Future studies are requiredto carefully characterize the responses of different color-responsive cell types to investigate this hypothesis. The dependence of predictability of inputs on the signal-to-noise ratio may potentially also explain the disappearance of gamma-band synchronization with prolonged visual stimulation (*Figure 6*, see next subsection).

## Dependence of gamma-band oscillations on hue and full-screen background

We demonstrated a prominent difference in gamma-band synchronization and firing activity between chromatic and achromatic surfaces. However, we also found prominent hue-related differences. These differences could reflect a constant property of the visual system to respond differently to particular hues, or might arise from other contextual processes, such as adaptation to the full-screen background (FSB) on which the surfaces were displayed. When using a gray FSB, we found that gamma oscillations were particularly strong for surface stimuli with red hues. This finding is consistent with previous work that used a gray FSB throughout and showed stronger gamma-band synchronization for red stimuli (*Shirhatti and Ray, 2018*; *Rols et al., 2001*). Note that with a gray FSB, gamma oscillations were reliably induced by all surface hues, which is consistent with our findings suggesting that both yellow-blue and red-green opponencies contribute to the generation of V1 gamma oscillations (*Figure 4—figure supplement 4*, *Figures 6* and *7*).

Importantly, we found that differences in gamma-oscillation strength among hues were highly dependent on the FSB and that with a black background a different dependence on hue emerged. In particular, with a black compared to a gray FSB, gamma oscillations increased in amplitude for green and yellow surface stimuli, but decreased in amplitude for red and blue surface stimuli. The quantitative model presented in *Figure 7* suggests that the M-cone pathway adapts more strongly than the L- and S-cone pathways, even when the adaptation-inducing FSB is supposedly 'neutral' (gray). By extension, background hues during natural vision would play a similar adapting role. An explanation for this phenomenon may be that, in general, uniform surfaces induce stronger or faster adaptation of the M-cone than L- and S-cone pathways, which may have a retinal, thalamic and/or cortical source. This interpretation is consistent with our finding that gamma-band amplitude decreased on a time-scale of seconds more rapidly for green than red or blue surfaces (*Figure 5*). We observed several other unique response features for green surfaces consistent with the idea of differential M-cone adaptation: First, we found that gamma oscillations were strongly dependent on luminance for green surfaces in particular, which suggests that a stronger luminance is needed to overcome adaptation (*Figure 4B*). Second, we found that gamma oscillations had a significantly lower peak frequency for green than for red or blue surface stimuli (*Figure 4B*). This suggests a weaker stimulus drive for green than red surface stimuli, because enhancing stimulus drive has been shown to increase the frequency of gamma oscillations (*Ray and Maunsell, 2010*; *Hadjipapas et al., 2015*; *Roberts et al., 2013*; *Henrie and Shapley, 2005*; *Jia et al., 2013b*). Third, with a gray FSB, we found that evoked MU transients were weaker for green than for equiluminant red and blue surface stimuli, which is consistent with increased adaptation of the M-cone pathway. Yet, we found that green stimuli exhibited a weaker decrease in firing over time (*Figure 4D*), and that the decrease in firing over time was particularly pronounced for red stimuli.

These findings have two important implications: (1) A gray FSB may differentially change neuronal responses to surfaces of different color hues. This may have implications for the design of studies examining differences in neuronal or behavioral responses between color hues. (2) Differential adaptation to distinct hues may have consequences for color perception in general. Interestingly, at the psychophysical level, it has been shown that psychophysical after-effects emerge more rapidly after viewing green than viewing red stimuli (*Werner et al., 2000*). Differential adaptation of M-cones may reflect important behavioral requirements of primates. The visual environment of primates is dominated by green and yellowish stimuli like leaves and trees (*Webster et al., 2007*). Detection of fruits with relatively high energy in red hues may be an important behavioral task for many primates (*Melin et al., 2017*). Trichomacy provides behavioral advantages for such detection (*Melin et al., 2017*), which may be aided by fast adaptation of M-cone derived signals.

## Differential effects of spatial and temporal context on gamma-band synchronization

This study reports two main findings: When the RF stimulus is part of a larger uniform surface, the spatial context allows a prediction of RF content, and gamma oscillations are enhanced. Yet, when the RF stimulus is part of a longer uniform stimulation period, the temporal context allows a prediction of RF content, and gamma oscillations are reduced. This suggests that the two effects are brought about by different mechanisms. When RF content is spatially predictable, enhanced gamma oscillations are accompanied by reduced firing rates. This pattern of results is highly suggestive of enhanced inhibition (*Vinck and Bosman, 2016*), and is in line with the prominent role of inhibition in the generation of gamma (see *Mechanisms of gamma-band synchronization* further below). By contrast, when the RF content is temporally predictable, the pattern of results is more suggestive of an adaptation mechanism leading to a progressive reduction in gamma strength. This part of our results is in line with previous reports from crossmodal and auditory studies (*Arnal et al., 2011*; *Todorovic et al., 2011*), which found enhanced gamma oscillations for unexpected stimuli. These findings are consistent with an earlier hypothesis, which stated that gamma oscillations should be enhanced for stimuli that generate prediction errors (*Bastos et al., 2012*). Note that the effects of both spatial and temporal predictability do not necessarily rely on top-down feedback.

The relationship of V1 gamma-band synchronization to temporal context may be more complex than suggested by this general conceptual notion, however: Previous studies have shown that when continuous stimulus motion has a large degree of jitter/randomness, either in case of entire video frames or in case of bar stimuli, V1-gamma-band synchronization tends to be weak, whereas V1 gamma-band synchronization tends to be strong in cases where stimulus motion is predictable (*Kayser et al., 2003*; *Kruse and Eckhorn, 1996*; *Vinck and Bosman, 2016*). Thus, the notion that V1 gamma-band synchronization increases when stimuli are unexpected or salient given the temporal context might apply only to discrete stimulus onsets and not generalize to cases where there is continuous stimulus motion. Furthermore, stimulus repetition can lead to a monotonic increase in V1 gamma-band synchronization over trials (*Brunet et al., 2014*). This increase of V1 gamma-band synchronization with stimulus repetition may reflect a slower learning process in which spatial center-surround interactions are modified by experience (*Vinck and Bosman, 2016*).

## Mechanisms of gamma-band synchronization

The results discussed above revealed several principles underlying the stimulus dependence of gamma synchronization. Yet, it remains unclear what precise neuronal mechanisms account for the emergence of V1 gamma synchronization, and its dependence on center-surround predictability. Previous work indicates that in primate and cat V1, gamma-band oscillations are generated cortically, specifically in the superficial layers of the cortex as well as layer 4B (*Xing et al., 2012*; *Buffalo et al., 2011*; *Livingstone, 1996*; *Herculano-Houzel et al., 1999*) . Furthermore, they have not been detected in the LGN of awake primates (*Bastos et al., 2014*). Together with the results presented in this paper, this indicates that the emergence of gamma oscillations in superficial layers depends on the integration of bottom-up inputs from the LGN and layer four with contextual information mediated through lateral and top-down feedback. Notably, superficial layers exhibit strong lateral connectivity and are densely innervated by top-down feedback (*Lund et al., 1993*; *Barone et al., 2000*; *Markov et al., 2014*). Within the cortex, the interaction between inhibitory and

excitatory neurons likely plays a critical role (*Jadi and Sejnowski, 2014*; *Veit et al., 2017*; *Hasenstaub et al., 2005*; *Buzsáki et al., 2012*; *Kopell et al., 2000*; *Whittington et al., 1995*; *Tiesinga and Sejnowski, 2009*; *Perrenoud et al., 2016*; *Cardin et al., 2009*; *Bartos et al., 2007*; *Sohal et al., 2009*; *Womelsdorf et al., 2014*; *Vinck et al., 2013b*). Specialized electrophysiological sub-classes of pyramidal neurons like chattering (fast-rhythmic bursting) cells, which have resonant properties in the gamma-frequency band, could also be a critical component of gamma rhythmogenesis (*Gray and McCormick, 1996*; *Cardin et al., 2005*; *Nowak et al., 2003*). Tangential, excitatory connections linking preferentially columns with similar feature preferences (e.g. color or orientation) may play a crucial role in synchronizing neuronal assemblies coding for related features (*Vinck and Bosman, 2016*; *Gray et al., 1989*; *Korndörfer et al., 2017*). A stimulus with high spatial predictability is likely to simultaneously activatea large number of preferentially coupled columns. This could then give rise to enhanced cooperativity among these columns and boost gamma synchronization by the recruitment of local excitatory and inhibitory neurons. In addition, feedback from higher visual areas could be critical, considering that the spatial spread of tangential connections is somewhat limited and covers a smaller surround region than cortical feedback (*Angelucci et al., 2017*).

## Functions of gamma-band synchronization

We finish with a discussion of the functional implications of the present findings. Early theories of gamma synchronization proposed that it may contribute to solving the 'binding problem' (*Singer and Gray, 1995*; *Singer, 1999*). This refers to the problem that the visual system segments images into segregated objects, which raises the problem that the local features comprising the object must at some processing stage be bound together. It was proposed that the activity of distributed neurons can be dynamically grouped together through synchrony according to perceptual Gestalt principles (*Milner, 1974*; *Engel et al., 1992*; *Singer and Gray, 1995*; *Singer, 1999*; *Von Der Malsburg, 1994*; *Singer, 2018*). Notably, functions that have been linked to surround modulation, such as contour integration (*Liang et al., 2017*), perceptual filling-in (*Zweig et al., 2015*; *Land, 1959*; *Wachtler et al., 2003*), and figure-ground segregation (*Lamme, 1995*), may contribute to perceptual grouping and underlie some of the Gestalt principles. Later work emphasized that gamma synchronization can flexibly regulate communication between neuronal populations (*Fries, 2005*; *Salinas and Sejnowski, 2001*; *Palmigiano et al., 2017*; *Akam and Kullmann, 2010*; *Knoblich et al., 2010*; *Colgin et al., 2009*; *Jia et al., 2013a*). For example, the communication-through-coherence hypothesis states that communication between neuronal populations can be flexibly modulated by selective coherence according to cognitive demands (*Friston, 2005*; *Fries, 2015*). Recent studies have shown that neuronal groups in distant visual areas show gamma-band coherence primarily when they processes an attended stimulus and that the level of coherence predicts behavioral benefits of attention (*Gregoriou et al., 2009*; *Bosman et al., 2012*; *Grothe et al., 2012*; *Rohenkohl et al., 2018*; *Buschman and Miller, 2007*).

In the context of efficient and predictive coding and the relationship of V1 gamma with spatial predictability, V1 gamma synchronization may play two functional roles (*Vinck and Bosman, 2016*), which remain to be tested:

1. Gamma synchronization may be a mechanism to increase the effective synaptic gain of V1 neurons on post-synaptic targets (e.g. V2) (*Salinas and Sejnowski, 2001*; *Fries, 2005*; *König et al., 1996*; *Bernander et al., 1991*; *Softky, 1994*; *Salinas and Sejnowski, 2000*) when a stimulus is efficiently encoded. This may ensure reliable transmission of V1 outputs even when firing is sparse, which is especially important in the presence of noise within or competing inputs to the receiving area.
2. Gamma synchronization could play an important role in coordinating the interactions between distributed V1 columns receiving related, and thereby redundant, visual inputs. The outputs of these columns need to be synaptically integrated, for which gamma synchronization could be a mechanism (*König et al., 1996*, *Fries, 2005*).

In sum, the present work provides evidence that visual cortex shows sparse and gamma-synchronized responses when surround stimulation predicts RF center stimulation. In contrast, firing rates are high when the surround does not predict the center. These effects are particularly pronounced in case of chromatic, compared to achromatic surfaces. A second key insight is that the FSB on which

surfaces are displayed strongly modulates gamma synchronization, in a way that suggests that uniform surfaces lead to stronger adaptation of the M-cone compared to L-cone pathways. This not only explains differences in gamma-band oscillations between surfaces of different hues, but may also have important behavioral and perceptual consequences, which needs to be explored in future work.

## Materials and methods

All procedures complied with the German and European regulations for the protection of animals and were approved by the regional authority (Regierungspräsidium Darmstadt).

### Surgical procedures

Two male adult macaque monkeys (Macaca mulatta) were used in this study (age 9–10 years, 15–17 kg). All surgeries for implantations were performed under general anesthesia and were followed by analgesic treatment post-operatively. A head post was implanted in both monkeys to allow for head fixation. In monkey H, we implanted CerePort ("Utah") arrays with 64 microelectrodes (inter-electrode distance 400 $\mu$m, tip radius 3–5 $\mu$m, impedances 70–800 kOhm at 1000 kHz, half of them with a length of 1 mm and half with a length of 0.6 mm, Blackrock Microsystems). One such array was implanted into area V1, another one in V4, both in the left hemisphere. The V4 array is not considered here. For array implantation, a large trepanation covering both areas was performed, the dura was cut open and reflected, arrays were inserted using a pneumatic device (Blackrock Microsystems), and both dura and bone were surgically closed. A reference wire was inserted under the dura toward parietal cortex. In monkey A, we implanted a semi-chronic microelectrode array Microdrive into area V1 of the left hemisphere (SC32-1, Gray Matter Research, containing 32 independently movable Alpha Omega glass insulated Tungsten electrodes with an impedance range of 0.5–2 MegaOhm and an inter-electrode distance of 1.5 mm). The microdrive chamber was used as the reference during recordings. The precise layers/depths that were recorded from could not be identified based on histological verification, which is the current gold-standard, because the animals are still alive. However, based on the observation that all sites in monkey H and the vast majority of sites in monkey A do not show the typical inversion of the event-related potential as is found in the deep layers (*Li et al., 2015*), we estimate that our recordings mainly sample activity from layers 2–4. Sites in monkey A and monkey H behaved qualitatively in a consistent manner across depths, such that all recording sites were pooled.

### Behavioral task

Both monkeys were trained on a fixation task. Monkeys were seated in a custom-made primate chair in a darkened booth. The two animals were positioned 83 (monkey H) or 64 cm (monkey A) in front of a 22 inch 120 Hz LCD monitor (Samsung 2233RZ, *Ghodrati et al., 2015*; *Wang and Nikolić, 2011*). Both monkeys self-initiated trials by fixating on a small fixation spot, which was presented at the screen center. Monkey H performed a pure fixation task. For monkey H, the fixation spot was a Gaussian with a white center, tapering smoothly into the background. For recordings with white background, the fixation spot color was changed to red. Note that the pattern of results for gray and white FSBs was very similar despite this difference (*Figure 6—figure supplement 2*), and that receptive fields were not covering the fovea. The task of monkey A was to report a change in the fixation spot from red to green or blue (randomly) with a lever release. The change in the fixation spot occurred only after the stimulus period and an additional 700 ms of background stimulation, during which the animal maintained fixation. For the recordings with colored backgrounds in monkey A, fixation colors were changed to remain visible, with a magenta fixation spot during the baseline and stimulus period. For both animals, trials during which the eye position deviated from the fixation spot by more than 0.8–1.5 visual deg radius were aborted. Correctly performed trials were rewarded with diluted fruit juice delivered with a solenoid valve system.

### Recordings

Data acquisition was performed using Tucker Davis Technologies (TDT) systems. Data were filtered between 0.35 and 7500 Hz (3 dB filter cutoffs) and digitized at 24.4140625 kHz (TDT PZ2 preamplifier). Stimulus onsets were recorded with a custom-made photodiode. Eye movements and pupil

size were recorded at 1000 Hz using an Eyelink 1000 system (Eyelink Inc.) with infrared illumination. Eye signals were calibrated before each recording session using a standardized fixation task. Behavioral control and stimulus presentation was done using in house custom software running in Matlab, including ARCADE (*Dowdall et al., 2018*).

## Visual stimulation paradigms during recordings

For all paradigms, stimuli were circular, did not have overlap with the fixation spot, and typically spanned a region from ca. 3–9 deg of eccentricity (monkey H) or 2.5–8.5 deg (monkey A, maximum: 1.6–9.6 deg for *Dataset 4*) in the lower right visual quadrant, matching RF locations. Trials always started with a baseline that lasted 0.5–0.6 s (monkey H) or 0.5–0.8 s (monkey A), and during which only the FSB and the fixation spot was shown. We used the following stimulus paradigms:

*Dataset 1*: For *Figures 1*, *4B* and *5*, we presented large uniform stimuli of 6 deg visual angle diameter on a gray FSB. For the chromatic condition, we used stimuli that were either green, red, or blue, at three different luminance levels (which are shown in *Figure 4*). For *Figure 1*, only the chromatic conditions with the highest available luminance level were used, approximately corresponding to the maximum possible luminance level for the blue primary. For the achromatic condition, we used either black (minimum luminance) or white (maximum luminance) stimuli.

The background was of an intermediate gray value that allowed for good eye tracking quality (see Table S1 for all luminance, RGB and CIE values). Stimulus duration was 3.3 s. This dataset included three sessions from monkey H and two sessions from monkey A. There were 20 $\pm$0 (H) and 20 $\pm$0 (A) trials in each session for each of the 11 conditions (two color hues * three luminance levels + black and white).

*Dataset 2*: For *Figure 2*, that is the size tuning paradigm, we presented a smaller (either 0.5, 1, or 2 deg) stimulus and a larger (6 deg) surface stimulus in the same trial sequentially, with each stimulus presented for only 0.6 s. In each trial, either the smaller ('small-first') or largest ('large-first') surface was presented first. In addition, we used an 'edge' condition in which the selected multi-unit's RF was centeredaround the vertical edge of the 6 deg stimulus, again followed or preceded by the standard full condition (*Figure 3—figure supplement 1*). The colors used were red, blue and green (at the same luminance intensities shown in *Figure 1*), black and white, and in case of monkey H, also orange, cyan and magenta hues. This dataset included five sessions from monkey H and four sessions from monkey A. There were 12.78$\pm$4.3 (H, 64 conditions) and 12.86$\pm$5.4 (A, 40 conditions) trials in each session for each of the conditions (four stimulus sizes * two presentation orders * 8/5 colors (H/A)).

*Dataset 3*: For *Figure 3*, we used only red, green and blue hues (with the same luminances as the maximum luminant red, green and blue used in *Dataset 1*, *Figure 1*). We presented three stimulus conditions: The uniform surface, the 'annulus' and the 'blob' condition (*Figure 3*). Stimuli in annulus or blob conditions were of the same size as the uniform surface, but the center 1 deg of the surface was either surrounded by a thin (0.25 deg) annulus of one of the other, equiluminant, hues, or filled completely with one of the other hues (*Figure 3*). For each surface of a given hue, there were therefore two 'annulus' and 'blob' conditions with the two remaining colors (*Figure 3*). In the analysis, we averaged over all the color combinations for a given condition, and compared the three main conditions. For monkey H, we additionally recorded two sessions with maximally luminant instead of equiluminant hues. Note that this generated strong luminance contrast changes between the colors, but yielded qualitatively similar results. This indicates that the observed effects do not depend on equiluminance, a condition that may occur rarely in nature. Because results were qualitatively similar, we pooled these sessions with the remaining five sessions of this animal. We used stimulus presentation times of 1.3–3.3 s. The first 1.3 s were analyzed, as in *Figure 1*. This dataset included seven sessions from monkey H and one session from monkey A. There were 15.88$\pm$0.21 (H) and 18.87$\pm$0.34 (A) trials in each session for each of the 15 conditions (three uniform conditions + 3 color hues * two color hues for mismatch * 2 (annulus vs blob mismatch)).

*Dataset 4*: For *Figure 4A*, we recorded 'rainbow' sessions in which surfaces (again 6 deg diameter size) of different colors were presented at the maximum possible luminance. We sampled the visible light spectrum linearly in 15 steps of equal size in terms of wavelength, with the MATLAB (MathWorks, Inc.) internal function spectrumRGB.m. Note that the monitor cannot produce line spectra, but can only approximate the corresponding hues through mixing of RGB channels (see e.g. *Figure 7A* for a yellow hue). We additionally included brown and pink (extra-spectral) hues (see

Table S1 and Figure *Figure 4—figure supplement 1*) and achromatic stimuli. This dataset included three sessions from monkey H and two sessions from monkey A. There were 12.1±6.9 (H) and 20±0 (A) trials in each session for each of the 22 conditions.For the analyses shown in *Figure 4B*, we used *Dataset 1*.

*Dataset 5*: For *Figures 6* and *7*, we used FSBs of various hues. The backgrounds used were red, green, blue and yellow at maximum possible luminance, as well as black, white and gray, presented at the same luminance intensities as in the other datasets. Surface stimuli of 6 (monkey H) or 8 (monkey A) deg diameter in size were used. The size was slightly increased for monkey A to place the edge of the surface stimulus further from the most peripheral RFs. The hues used for the surface were identical to the ones used for the FSBs. In addition, we presented chromatic surfaces with reduced values, namely red, green and blue with the same luminance levels as in *Figure 1*, and a brown surface. All possible combinations of surface and FSB hues were shown. All other presentation parameters were kept as for *Dataset 1*. This dataset included 16 sessions from monkey H and nine sessions from monkey A (1–2 per FSB). There were 17.89±0.17 (H) and 19.00±0.08 (A) trials in each session for each of the 11 conditions.

For all stimulus paradigms for monkey A, and in *Dataset 5* for monkey H, there was a post-stimulus period of 0.7 s (0.5 s in monkey H) after the offset of the stimulus, during which the monkeys was required to maintain fixation. For monkey A, the fixation color would change after this period and the monkey had to respond to this change with the release of a lever, whereupon the fixation spot was removed. Presentation of different stimulus conditions was in a pseudo-random order.

## DKL color space

In order to calibrate the monitor outputs, the luminance of the RGB monitor primaries were measured with Konica Minolta CS-100A chroma meter and look-up tables were generated. Monitors were gamma-corrected to linearize the dependence of luminance on RGB values.

The Derrington-Krauskopf-Lennie (DKL) Color Space was introduced as a color-opponent modulation space (*Krauskopf et al., 1982*; *Derrington et al., 1984*). DKL color space is based on a cone-contrast representation, where cone activation to a color stimulus is quantified as the relative change of the cone activations with respect to the background color (*Brainard, 1996*). Weber cone-contrasts are computed in three steps: (1) The change in cone-activation relative to the full-screen background is computed, (2) This change in cone-activation is normalized (divisively) by the extent to which the background differentially activates the different cones (*Brainard, 1996*). These cone contrasts are then transformed into three primary axes of the DKL space, which correspond to the mechanisms of L + M (luminance), LM (red-green opponency), and S-(L + M) (blue-yellow opponency). Along the L-M axis, maximum L/M cone contrasts were 9.60% and 14.81% respectively, along the S-(L + M) axis S cone contrast was 79.35%. These values were found to be similar to previous studies (*Hansen and Gegenfurtner, 2013*; *De Valois et al., 2000*).

## Data analysis
### Preprocessing

Data were analysed in MATLAB using the FieldTrip toolbox (*Oostenveld et al., 2011*). Only correctly performed trials were analyzed. LFPs were derived from the broadband signal using MATLAB's decimate.m function, by low-pass filtering with a cutoff frequency of 24414.0625/24/2 Hz (FIR Filter with order 30) and downsampling to 24414.0625/24 Hz. Line noise was removed using two-pass 4th order Butterworth bandstop filters between 49.9–50.1, 99.7–100.3 and 149.5–150.5 Hz. LFPs had a unipolar reference scheme described in *Recordings*. Explorative analyses with local bipolar derivations, obtained by subtracting the signals from immediately neighboring electrodes from each other, yielded comparable results (data not shown). MU signals were derived from the broadband signal through bandpass filtering between 300 and 6000 Hz (4th order butterworth), rectification, and applying low-pass filtering and downsampling the same way as for the LFPs. For the calculation of rate modulations, this MU signal was smoothed with a Gaussian kernel with an SD of 20 ms. Qualitatively similar results were obtained using thresholded multi-unit data. We used this MU signal for all analyses in the main text, as in previous studies by other labs (*Schmid et al., 2013*; *Self et al., 2013*; *Xing et al., 2012*; *Legatt et al., 1980*).

## Receptive field estimation

Receptive fields were mapped with moving bar stimuli (spanning the entire monitor). Moving bars (width 0.1 deg, speed 10/17 deg/s) were presented in eight orientations for monkey H and 8–16 orientations for monkey A, each for 10–20 repetitions. Mapping sessions were intermittent for monkey H and typically daily for monkey A, to confirm stability of the recordings. MU responses were projected onto the stimulus screen, after shift-correction by the response latency that maximized the back-projected response. MU responses were then fitted by a Gaussian function. This Gaussian was used to extract the 10th percentile and the 90th percentile, and this was done separately for each movement direction. Across the 16 directions, this yielded 32 data points, which were fit with an ellipse. This ellipse was defined as that MU's RF. The RF size is defined as the diameter based on (area of the ellipse/pi)*2.

## Electrode selection

We included all electrodes for analysis that met the following criteria: the MU showed a response to RF stimulation that was at least two SDs above stimulation outside the RF . The MU response during the response period (0.05–0.15 s) of at least one condition of the respective dataset was at least 2 SD above the corresponding baseline (−0.1–0 s). In case of *Figures 2–3*, it was additionally required that the RF center of the MU was within 0.5 deg of the stimulus center. In the remaining figures, it was required that the RF center was within the surface stimulus.

## Estimation of LFP power spectra

For *Figures 1*, *3–4* and *6–7*, the baseline period was the last 500 ms before stimulus onset, and each stimulation period yielded two non-overlapping epochs of 500 ms (0.3–1.3 s period). For *Figure 2*, due to the short presentation times, we used epochs of 300 ms (300–600 ms after the onset of the stimulus, and for baseline 300 ms before stimulus onset). LFP epochs were multiplied with discrete prolate spheroidal sequences (multi-tapers for ±5 Hz smoothing), Fourier transformed and squared to obtain LFP power spectral densities (for a recent discussion on spectral estimation see *Pesaran et al., 2018*). For *Figure 5*, we used windows of 0.3 s length, slid over the data in steps of 50 ms. Data were multiplied with a Hann taper before Fourier transformation.

## Normalization of LFP power spectra

To show LFP power changes, we computed relative power spectra by dividing single-trial power spectra from the stimulation period by the average power spectra across conditions and trials from the baseline. This was shown as a fold-change in all figures showing relative changes except for *Figure 5* TFRs, where for visualization purposes, we transformed this into dB units.

To investigate absolute LFP power (without reference to the baseline), we normalized power spectra per electrode by the total power above 25 Hz in the baseline condition. This normalization reduced variance or scaling in the LFP power spectra across sessions and animals before averaging. By normalizing both the baseline and the stimulus period by the same normalization factor, we could still examine changes in raw LFP power across conditions, for each frequency bin separately. This would not have been possible if we had normalized the LFP power spectrum in a given condition by the total power across frequencies in the same condition. These power spectra were averaged across the selected channels (except for single-channel analyses as in *Figure 1—figure supplement 2*).

## Quantification of LFP gamma-band amplitude

Quantification of the differences in gamma-band amplitude between conditions is in general a difficult problem because changes in firing rate can cause broad-band shifts in the LFP power spectrum, and because spikes can 'bleed-in' at higher LFP frequencies (*Miller et al., 2009*; *Ray and Maunsell, 2011*; *Pesaran et al., 2018*; *Buzsáki et al., 2012*). We developed an algorithm to extract gamma-band amplitude in order to address these problems (see *Figure 1—figure supplement 1*, for an illustration). We present two versions of this algorithm that are used for separate figures, and are based on constructing a polynomial fit of the LFP spectrum which was detrended in two separate ways. The first algorithm had the following structure:

1. Power spectra were log-transformed and the frequency axis was also sampled in log-spaced units to avoid over-fitting of high-frequency datapoints. All subsequent polynomial fits were performed on the 20–140 Hz range.
2. We used the change in stimulus-induced LFP power versus the common baseline (see above), expressed as $\Delta P = \log(P_{stim}) - \log(P_{base})$
3. To determine the polynomial order, we used a cross validation procedure to prevent overfitting. A random half of the trials was used for the fitting and deemed the 'training set'. The remaining trials were the 'test set'. Polynomials of order 1–20 were fit to $\Delta P$ as a function of frequency for the 'training set', minimizing the mean squared error. We then computed the mean squared error using the same polynomial fit on the 'test set' for each of the 20 orders. This procedure was then repeated for multiple (50) iterations, with a random half of the trials selected for each iteration, and for each iteration, the best-performing order was retained.
4. A polynomial with the median of the best-performing orders was then fit to the complete set of trials.
5. On the polynomial fit, local maxima and minima in the 30–80 Hz range were identified. The peak gamma frequency was the location of the maximum. The band-width of gamma was estimated as twice the distance between the frequency of the maximum ($F_{max}$) and the frequency of the first local minimum to the left of the maximum ($F_{min}$), that is $b = 2F_{max} - F_{min}$ (neuronal). The gamma amplitude was then assessed from the difference between the value of the polynomial fit at the maximum and the average of the polynomial fit at $F_{min}$ and $F_{max} + F_{min}$.
6. This difference was taken in log-space (because the power spectra were originally log-transformed) and then transformed to a fold-change.

If firing rate changes relative to baseline (or between conditions) were very strong, for example with small stimuli, this fitting procedure occasionally ran into problems, because relative LFP power spectra showed broad increases that were likely due to non-rhythmic processes like spikes or postsynaptic potentials (see *Figure 3* for an example of this effect). In addition, in *Figures 6* and *7*, because we used background stimuli of different hues, a 'neutral baseline' like the gray background screen was not always available. In these cases, we modified the second step of this algorithm. Instead of computing the change in LFP power relative to baseline, we performed a $1/F^n$ correction on the raw LFP power spectrum. The $1/F^n$ correction was performed by fitting an exponential to the LFP power spectrum, excluding data points in the typical gamma range of 30–80 Hz. Note that we fitted an exponential function because in many cases, bleed-in of spiking energy in the LFP caused a departure from a linearity in the log(power) versus log(frequency) graph (see also *Haller et al., 2018*; *Shirhatti and Ray, 2018*). We visually inspected the fits for a large number of spectra and compared this also to a procedure with a mixture of a linear fit and a Gaussian fit to the log(power) versus log(frequency) graph, which had substantially more problems in dealing with spike-bleed at high frequencies, as well as with additional peaks (potentially harmonics) at higher frequencies (e.g. for the red surfaces) (data not shown).

## Spike-field coherence

For spike-field coherence, we used only electrodes selected by the procedure described above. In addition, for LFP-MUA pairs, we required that the electrodes were direct neighbors in the grid, and in the case of monkey H, given that the microelectrode array had two fixed depths, were of the same depth. Spike-field phase-locking was computed as follows. We estimated the cross-spectral density between LFP and MU signal for each trial separately (cross-spectra) using the same spectral estimation settings as for the LFP power spectrum. This yielded one cross-spectrum per trial. We then normalized the cross-spectrum per trial by its absolute values, to obtain the cross-spectral phases (without amplitude information). We used those normalized cross-spectra to compute the Pairwise Phase Consistency (PPC), using FieldTrip (*Oostenveld et al., 2011*). This measure has the advantage that the bias by trial count, inherent tofor example the spectral coherence, is avoided (*Vinck et al., 2010b*). For a given MU site, the PPC values were then averaged across all the combinations with LFPs from the other selected channels. Note that MU-LFP combinations from the same electrode were excluded to avoid artifactual coherence due to bleed-in of spikes into the LFP (*Ray and Maunsell, 2011*; *Buzsáki et al., 2012*). Because of the distance between electrodes (at least 400 micrometer), this was not an issue for MU-LFP combinations from different electrodes.

The standard error of the PPC was estimated across sessions. This was different from SE estimation for power and rate, which used the bootstrap (see below). Bootstrap estimates are problematic

for PPC because bootstraps contain repetitions of identical trials, which trivially yield high coherence values.

## Rate modulation

Rate modulation was computed as $\log_{10} M_{stim}/M_{base}$, where $M_{stim}$ and $M_{base}$ represent the MU firing activity in the stimulus and baseline period, respectively. To quantify surround suppression, we took the differences of these rate modulation indices between small and large stimulus size conditions.

## Modulation index of fold-changes

To quantify the modulation of LFP gamma-amplitude (expressed as fold-change) between conditions (*Figures 5* and *6*), we computed a modulation index as $(A - B)/(A + B)$, where $A$ and $B$ are the gamma-amplitudes in the two conditions, taken as the fold-change minus 1. Note that the fold-change was extracted using the polynomial fitting procedure described above, and a fold-change of 1 indicated the absence of a gamma peak.

## Microsaccade detection and subsequent LFP analysis

For microsaccade detection, we smoothed horizontal and vertical eye signals (rectangular window of $\pm 5$ ms) and differentiated the signals over time points separated by 10 ms to obtain robust eye velocity signals. For monkey H, for whom data from both eyes were available, data were averaged across eyes. We then used the microsaccade detection algorithm described in *Engbert and Kliegl (2003)* with a velocity threshold of 6*$c$, where $c$ is the criterion defined as $c \equiv \mathrm{Median}[v^2] - (\mathrm{Median}[v])^2$. Threshold crossings in either the horizontal or vertical direction were considered as microsaccades. We tested several threshold levels and obtained qualitatively similar results. We then removed data epochs of 100 ms after each microsaccade and recomputed our analyses (based on *Lowet et al., 2016*; 100 ms is approximately the duration of microsaccade effects in V1). Removing 200 ms after each microsaccade yielded qualitatively similar results but fewer remaining epochs. For the analysis of LFP gamma amplitude, we switched to analyzing epochs of 100 ms using a Hann taper, instead of the 500 ms time bins used before. This is sacrificing some frequency resolution and limiting the results to frequencies >20 Hz, in order to obtain a large number of microsaccade-free epochs. Epochs were zero-padded to 1 s, effectively smoothing the spectra. Note that we show the results for the data including microsaccades with the identical epoch length and taper to allow a fair comparison.

## Pupil responses

Pupil signals across the two eyes were averaged for monkey H. Pupil size during the comparatively stable period 200 ms to stimulus onset was used as a baseline. Pupil size was then computed as percent change from the average response during this time (A-B)/B, where A is the pupil response at each time point and B is the average response during the baseline period. Note that since the Eyelink system gives outputs with arbitrary units, and these were negative during the baseline period, we took the absolute value for the denominator such that pupil size decreases are indicated by negative values.

## Statistics

Error bars or shaded error regions correspond to $\pm$ one standard error of the mean (SEM). SEM was estimated using a bootstrap procedure, with the exception of spike-field coherence (see above). For the $b$-th bootstrap out of $B = 1000$ bootstraps, $b = 1, \ldots, B$, the following was done. For each condition in a given session, with a set of $N$ trials $\mathcal{T}$, we took a random set of $N$ trials from $\mathcal{T}$ with replacement, yielding a new set of trials $\mathcal{S}_b$. For that sample of $N$ trials $\mathcal{S}_b$, we then computed the statistic of interest. For LFP signals, we then computed the average statistic in a given session over all channels, then averaged over sessions, and then monkeys. The rationale behind averaging across all LFP channels was that these signals are likely highly statistically dependent because of volume conduction among the relatively closely spaced electrodes. For MU signals, we computed the average statistic of interest across sessions per MU site separately, and then averaged across all recording sites. The standard error of the mean was then defined as the standard deviation over the $B$ average statistics, as is common with bootstrapping procedures.

We used the bootstrap distributions for inference on fold-change estimates or fold-change modulation indices between conditions, as well as differences in peak gamma frequency. In this case, we computed for each bootstrap the difference between average statistics for two conditions, and then tested whether this distribution was different from zero (with Bonferonni correction for number of comparisons).

For frequency- or time-resolved differences (in absolute and relative LFP power spectra and rate modulation scores), we used multiple-comparison corrected permutation tests: In this case, we shuffled the trials between two conditions per permutation $P$ times, and then constructed a permutation distribution of average absolute differences between conditions. We equalized trial numbers for each comparison, for example between chromatic/achromatic conditions or the different stimulus sizes. We then compared the observed difference between average statistics against this permutation distribution. For multiple-comparison correction, we used the procedure from *Korn et al. (2004)*, which is based on the sorted distribution of absolute differences, with alpha and false discovery rate values of 0.05. In this iterative procedure, values in the observed distribution exceeding the 95th percentile of the $P$ maximal values of each permutation distribution (critical value) are deemed significant. Significant values are removed from the observed distribution, and the same positions are removed from all $P$ permutation distributions. Values in the observed distribution exceeding the critical value based on these permutation distributions are then iteratively collected until no value in the observed distribution exceeds the critical value. Note that statistical parameters are reported mostly in the figure captions.

## Quantitative model for dependence of gamma-band amplitude on background stimulus

Cone data were extracted from *Hárosi (1987)* (bleaching difference corrected spectra). Polynomials of order seven were fit to these curves. The cone response curves were then normalized to the maximum. We measured the spectral energy of each color as well as black, white and gray (Ocean Optics WaveGo; XWAVE-STS-VIS-RAD). The spectral energies of the colors were normalized to unit mass. For gray, we added the normalized energies of R, G and B and multiplied with the energy ratio of gray over white. We then convolved the cone response curves with the normalized spectral energies to determine how strongly each background adapts the three cones. Regression models were then fit as explained in the Results text and caption of *Figure 7*. SEM for regression coefficients are obtained by the same bootstrap procedure as described above.

## Acknowledgements

AP, CU and MV conceived of the idea of the study and designed the experiments. AP, CU, JKL and RR performed recordings. AP, JKL, RR, SS and WB performed initial behavioral training. JKL, SS, WB, and WS planned and performed surgical implants. For this we are also thankful to Michael Schmid and Richard Saunders. We also wish to acknowledge Emmy Noether 2806 to Michael Schmid. AP, JKL, JRD, WS and PF collected preliminary (unpublished) data on hue differences; in this context we would also like to acknowledge Gareth Bland and Marieke Schölvinck. AP, CU and MV performed data analysis. AP, CU, WS, PF and MV wrote the paper, with help from comments of the other authors. We would like to thank Quentin Perrenoud for very helpful comments.

## Additional information

### Funding

| Funder | Grant reference number | Author |
| --- | --- | --- |
| Deutsche Forschungsgemeinschaft | SPP 1665 | Pascal Fries |
| Deutsche Forschungsgemeinschaft | FOR 1847 | Pascal Fries |
| Deutsche Forschungsgemeinschaft | FR2557/5-1-CORNET | Pascal Fries |

| Deutsche Forschungsge-meinschaft | FR2557/6-1-NeuroTMR | Pascal Fries |
|---|---|---|
| Deutsche Forschungsge-meinschaft | Reinhart Kosselleck grant | Wolf Singer |
| National Institutes of Health | 1U54MH091657-WU-Minn-Consortium-HCP | Pascal Fries |
| European Science Foundation | European Young Investigator Award | Pascal Fries |
| LOEWE | NeFF | Pascal Fries |
| European Commission | HEALTH-F2-2008-200728-BrainSynch | Pascal Fries |
| European Commission | FP7-604102-HBP | Pascal Fries |

The funders had no role in study design, data collection and interpretation, or the decision to submit the work for publication.

## Author contributions

Alina Peter, Cem Uran, Conceptualization, Resources, Data curation, Software, Formal analysis, Validation, Investigation, Visualization, Methodology, Writing—original draft, Writing—review and editing; Johanna Klon-Lipok, Sylvia van Stijn, Investigation, Methodology; Rasmus Roese, Investigation; William Barnes, Investigation, Methodology, Writing—review and editing; Jarrod R Dowdall, Resources, Investigation, Writing—review and editing; Wolf Singer, Resources, Supervision, Funding acquisition, Writing—original draft, Writing—review and editing; Pascal Fries, Conceptualization, Resources, Supervision, Funding acquisition, Writing—original draft, Project administration, Writing—review and editing; Martin Vinck, Conceptualization, Resources, Software, Formal analysis, Supervision, Funding acquisition, Validation, Investigation, Visualization, Methodology, Writing—original draft, Project administration, Writing—review and editing

## Author ORCIDs

Alina Peter (iD) http://orcid.org/0000-0001-8497-6235
William Barnes (iD) http://orcid.org/0000-0002-3833-1214
Pascal Fries (iD) https://orcid.org/0000-0002-4270-1468
Martin Vinck (iD) http://orcid.org/0000-0002-4044-0970

## Ethics

Animal experimentation: All procedures complied with the German and European regulations for the protection of animals and were approved by the regional authority (Regierungspräsidium Darmstadt, F-149-1000/1005). All surgeries were performed under anesthesia and were followed by analgesic treatment post-operatively.

## Decision letter and Author response

Decision letter https://doi.org/10.7554/eLife.42101.027
Author response https://doi.org/10.7554/eLife.42101.028

## Additional files

### Supplementary files

• Supplementary file 1. RGB values, luminances (cd/m$^2$) and CIE values (*1000) used in this study. Luminances and CIE values were measured with a Konica Minolta CS-100A chromameter, CIE values refer to the 1931 two degree standard observer. Standard black, white and gray used across datasets are in rows 1–3.
DOI: https://doi.org/10.7554/eLife.42101.022

• Transparent reporting form
DOI: https://doi.org/10.7554/eLife.42101.023

## Data availability

As described in the Methods section, several datasets were acquired in this study. Datasets have been uploaded onto Dryad (https://doi.org/10.5061/dryad.4809qj4). These include individual sessions with each session preprocessed (downsampled, see Methods) with MUA and single-trial spectra as described on Dryad.

The following dataset was generated:

| Author(s) | Year | Dataset title | Dataset URL | Database and Identifier |
|---|---|---|---|---|
| Alina Peter, Cem Uran, Johanna Klon-Lipok, Rasmus Roese, Sylvia van Stijn | 2019 | Data from: Surface color and predictability determine contextual modulation of V1 firing and gamma oscillations | https://doi.org/10.5061/dryad.4809qj4 | Dryad Digital Repository, 10.5061/dryad.4809qj4 |

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
