## [Decision Letter]

Thank you for sending your article entitled "Surface color and predictability determine contextual modulation of macaque V1 firing and gamma oscillations" for peer review at *eLife*. Your article has been evaluated by Laura Colgin as the Senior Editor, a Reviewing Editor, and two reviewers.

Both reviewers felt that the study has many merits and potentially provides an important contribution to the field. But serious concerns were raised that require substantial revisions to the manuscript and thus preclude a simple revise and resubmit decision. For example, both reviewers expressed that differences in luminance/contrast/color attributes were not well controlled, and they felt that results might be explained by these differences (i.e. such differences could cause a stronger drive to cortical circuits, which in turn could cause the observed increases in gamma). Therefore, all agreed it would be best to provide this feedback and inquire as to whether and how major concerns could be addressed. The two reviews are provided in their entirety below. The action plan only needs to address the major comments. The action plan should contain specific details about planned revisions and a realistic timeline to complete the proposed efforts.

Reviewer #2:

The paper presents very strong differential effects of manipulation of color and luminance on the amplitude of gamma oscillations in monkey V1. The figures are very well produced, the writing is for the most partvery clear and the data seem to have been carefully collected and appropriately analysed. The basic phenomenology reported here is therefore clear, appears robust across the two monkeys, and is likely of high interest to both those who study gamma oscillations and those who study color.

I have two misgivings, somewhat related. First, the stimuli, up to the modelling component at the end, are characterisedbasically by what is displayed on the monitor. For a person who studies color this is irritating. It is very hard to guess (even with reference to the CIE values in the Table) how each of the stimuli are represented by the photoreceptors (most have combined luminance and chromatic variation, meaning they modulate each of the cone classes but in slightly different ways). It would help, I think, to have a Figure at the beginning of relevant sections that represents each of the stimuli in terms of cone contrast (L, M, S), e.g. in MB-DKL space.

Second, and more substantively we know that the retina and LGN will have very different population spiking activity for large chromatic and achromatic stimuli. The authors mention several times 'single-opponent' cells in LGN or V1. By single-opponent I think they mean 'Type II' in Hubel and Wiesel's typology, which have co-extensive cone-opponent subunits (e.g. S cone and LM cone subunits that have the same spatial profile, but opposite sign). But in most retinal/LGN cells, especially those in the parvocellular L-M pathway, and many V1 units in the inputs layers, have different spatial summation regions for different cone-inputs (Type I). The result is that for chromatic modulation these cells shows increasing response with increasing size, but for achromatic modulation response peaks at a small size and declines in larger sizes. The consequence is that the drive to cortical circuits will be stronger for large chromatic fields, and weaker for large achromatic fields. If gamma were simply stronger for stronger inputs (from LGN to V1, or from layer IV to other layers), this may explain many of the results. As the authors show, the effect of adaptation state on gamma can be largely explained by variation in retinal output, and similarly (e.g. Figure 2B) chromatic and achromatic LFP responses are similar for sizes up to 2 deg, and only diverge for much larger sizes (where the chromatic increases and the achromatic decreases). The achromatic and chromatic tuning of LGN output will show similar tuning. Note that several labs have shown that there are strong surrounds for V1 neurons (particularly in the upper layers) that are broadband in their color selectivity (e.g. Ts'o and Gilbert, 1988) which means that the MUA may not show the chromatic advantage for large fields that is present in the LGN and input layers. The current framing does not deal adequately with these possibilities – for example, in subsection “Dependence on full-screen background hue” the statement is made that the gamma is not a mere consequence of input drive to a larger cortical region. This conclusion needs substantial explication and needs consideration in terms of the above. I think however that my concern is not terminal – what I would like to see is if a reasonable model of retinal/LGN output would predict the spatio-chromatic dependence of the gamma power. This is simply an elaboration of the modelling effort already performed.

Reviewer #3:

The paper by Peter and Alina et al. studies how color and luminance surfaces are encoded by population of neurons in V1. The authors recorded and analyzed LFP and MUA in V1 while 2 monkeys were presented with chromatic or achromatic surface stimuli. The main result of the paper is an increase power the LFP gamma-band and decreased firing rate mainly for color surfaces (as opposed to luminance surfaces), when the RF of the MUA is positioned at the center of the large surface (6 deg). They varied the surface's color, size, surround and investigated the background adaptation. In addition, they developed a model to explain their observation and claim that the results support the predictability hypothesis (Vink and Bosman, 2016). This is a very extensive study with novel results that are potentially interesting and important however there are some major concerns as indicated below.

1) Although the number of sessions and trials per figure/analysis is reported, the reproducibility of the results across the two animals is unclear to me. Please clarify this issue: which results were reproduced for both animals? Please indicatefor each data set – how many sessions were obtained from each animal and the number of trials, for each animal. This is particularly important because the Methods' text indicates that the two animals performed two different behavioral tasks. Statistical significance tests – grand analysis: it is unclear whether the authors used the combined distributions of all single trials within a dataset. Is each single trial an independent measure for chromatic vs. achromatic conditions? Or did they use the mean (across trials) for each session and then tested the statistical significance across sessions? I am asking this because usually the reproducibility and statistical significance is tested across independent recording sessions (and not the single trials).

2) A central problem in this study is that the luminance/contrast/color attributes of the different stimuli were not well equated for the different conditions, which makes the interpretation very complex. This is based on the Materials and methods section and Table 1 in the Supplementary material. For example, when using chromatic stimuli the gray background (27.5 cd/m^2^) was not equal with luminance to the chromatic stimuli (Dataset 1: the color surfaces luminance varied mostly between 2.5 to 10 cd/m^2^). Thus, when color stimuli appeared over the screen there was a change also in the luminance level (compared with background). In addition, the white and black stimuli are not symmetric relative to the background. What about the luminance and L-M cone contrast: was the luminance weber contrast of the black/white stimuli adjusted to the L-M cone weber contrast of the chromatic stimuli, all relative to background? Please show the numbers. Note that Figure 4 does not provide answer for all these questions.

3) The authors indicate that the MU's RF were much smaller than the surfaces they have used, however I could not find the mean or median size of the RFs (as opposed to RF eccentricity that is clearly reported) for each dataset. Please report also what was the ratio between the large surface stimulus (6 deg) and the MUA RF size, for each data set.

4) Late decrease in firing rate and increase in gamma-band power: what is the relation to eye movement? The authors need to control for this. Can it be that the late decrease of firing rate towards/below baseline has to do with microsaccades/small saccades that the animals start to perform at times> 200 ms? which is typical to microsaccades/saccadic inhibition? In addition, surface stimuli of 6 degree, especially the white surface that is reported to be 127 cd/m^2^ (over a gray background of 27.5 cd/m^2^), can induce changes in pupil size. A previous paper by Haynes et al. (2004) found a change in the humans' pupil due to light variation of the surface stimuli. Can the authors account for this effect in the animals?

5) Many previous studies (papers from Shapley, Alonso and others) showed that V1 neurons have higher activity for black vs. white stimuli. Is there any difference in the measured neuronal results between white and black surface? Given this answer, was it justified to combine the black and white responses?

6) The neural correlates of color surfaces and in particular the increased power in the gamma-band -seems to appear late in time. Firing rates appear > 200 ms and gamma-band power is computed over 600-1.3 s intervals. If we assume that color processing occurs already with the first 200 ms, can the authors clarify, what information on color surfaces, these late modulations hold?

[Editors' note: further revisions were requestedprior to acceptance, as described below.]

Thank you for resubmitting your work entitled "Surface color and predictability determine contextual modulation of macaque V1 firing and gamma oscillations" for further consideration at *eLife*. Your revised article has been favorably evaluated by Laura Colgin (Senior Editor and Reviewing Editor) and two reviewers.

The manuscript has been improved but there are some remaining issues that were raised by the first reviewer that need to be addressed before acceptance, as outlined below:

Essential Revisions:

A few issues remain concerning the approach to addressing the main point (drive to cortex). In reply to major comment 2 the authors (point A and C4 of response) state that large chromatic surfaces do not lead to strong V1 firing rates but do yield strong gamma. If the authors mean that the firing rates in their study seem similar for chromatic and achromatic stimuli this seems correct, however there it likely that their unit activity recordings are biased away from layer 4, and there is also is reasonably good evidence that chromatic stimuli will yield stronger overall firing rates for uniform fields than will achromatic stimuli (see for review Schluppeck and Engel, (2002). I am not sure why the authors argue that chromatically varying center-surround stimuli will produce as strong a response from LGN (or V1) standard units (point (B) in response – they will certainly be less effective for all but double opponent cells, which are not present in LGN, and are rare in V1. The same goes for chromatic edges. While the points above may seem semantic they are still important – they mean that the authors have thoroughly ruled out some variant of 'drive' as the primary correlate of gamma in these experiments.

---

## [Author Response]

[…] Reviewer #2:The paper presents very strong differential effects of manipulation of color and luminance on the amplitude of gamma oscillations in monkey V1. The figures are very well produced, the writing is for the most partvery clear and the data seem to have been carefully collected and appropriately analysed. The basic phenomenology reported here is therefore clear, appears robust across the two monkeys, and is likely of high interest to both those who study gamma oscillations and those who study color.

We thank the reviewer for his/her comments. We have numbered the two major comments of the reviewer below.

I have two misgivings, somewhat related. First, the stimuli, up to the modelling component at the end, are characterisedbasically by what is displayed on the monitor. For a person who studies color this is irritating. It is very hard to guess (even with reference to the CIE values in the Table) how each of the stimuli are represented by the photoreceptors (most have combined luminance and chromatic variation, meaning they modulate each of the cone classes but in slightly different ways). It would help, I think, to have a Figure at the beginning of relevant sections that represents each of the stimuli in terms of cone contrast (L,M,S), e.g. in MB-DKL space.

We followed the request of the reviewer and added the DKL space for the relevant figures (Figure 4—figure supplement 1, Figure 4—figure supplement 2, Figure 4—figure supplement 4, and Figure 6—figure supplement 1 – see end of this reviewer’s response.)

Second, and more substantively we know that the retina and LGN will have very different population spiking activity for large chromatic and achromatic stimuli. The authors mention several times 'single-opponent' cells in LGN or V1. […] I think however that my concern is not terminal – what I would like to see is if a reasonable model of retinal/LGN output would predict the spatio-chromatic dependence of the gamma power. This is simply an elaboration of the modelling effort already performed.

We thank the reviewer for these useful comments. In response, we have performed extensive additional analyses and included several new figures and corresponding text. The main question of the reviewer is whether the drive is the key determinant of the amplitude of V1 gamma-band oscillations, and of the difference in gamma-band amplitude between chromatic and achromatic stimuli. We address these points as follows:

A) If gamma reflects chromaticity-related drive from LGN, the specific translation of this drive into cortical gamma is a novel and important insight.

We agree with the reviewer that large chromatic surfaces lead to strong input drive from the LGN. We note that this drive from LGN does not translate to strong firing rates in V1. By contrast, itdoes lead to strong gamma. This specific translation of chromaticity-related drive into cortical gamma, rather than into firing rates or power at other frequencies, is a novel and important insight. It constitutes one of the main scientific advances of this manuscript.

Beyond this insight, the data show that chromaticity-related drive alone cannot fully explain visual cortical gamma-band activity, as explained in (B).

B) Strong gamma requires chromaticity and predictability.

The data show that strong visual cortical gamma requires that stimuli are not only chromatic, but also predictable. This is prominently visible in Figure 3 and Figure 3—figure supplement 1.

1) Figure 3 shows that large chromatic surfaces, in which the center cannot be predicted (due to center mismatch or annulus) lead to strongly reduced gamma. Note that such surfaces still provide strong drive from LGN. Note also that they induce much stronger firing rates in cortex than homogeneous surfaces.

2) Predictability is similarly violated at the edge of a surface, and Figure 3—figure supplement 1 shows that this has similar effects on gamma and firing rates.

We have revised the Discussion section to convey these points:

“Overall, our data suggests that both sufficient drive and spatial predictability are the necessary ingredients for the generation of V1 gamma. There are several cases in our manuscript where differences in V1 gamma are dissociated from differences in firing rate. For example, in Figure 3 we show a strong increase in firing rates for the annulus and blob condition as compared to the uniform surface, however gamma oscillations are markedly decreased. In Figure 3—figure supplement 1B, we show that gamma-band oscillations are stronger at the center than the edge of a chromatic surface stimulus, however firing rates are much stronger at the edge of the chromatic surface stimulus. On the other hand, we also present cases where stimuli have high spatial predictability, but where gamma is weak or absent likely due to low drive (Figure 1 achromatic stimuli, Figure 6 chromatic stimuli after prolonged adaptation).”

An important question is how to functionally relate the dependence of gamma on both predictability and drive.

We furthermore explored how chromatic drive relates quantitatively to gamma power, as explained in (C).

C) Modeling cortical gamma power.

We followed the request of the reviewer and explored whether cortical gamma can be modeled by retinal/LGN output, estimated based on different cone-contrasts.

We first explored in further detail how gamma-band power relates to DKL cone-contrast for the set of chromatic stimuli (points C1 and C2 below). By considering achromatic stimuli on chromatic backgrounds, we also considered how gamma for achromatic stimuli depends on DKL cone-contrast along the L-M and S-(L+M) direction (point C3 below).

C1) Correlation between DKL cone-contrast and gamma:

We agree with the reviewer that the difference in gamma between chromatic and achromatic stimuli is likely due to the absence of cone-contrast in the L-M and S-(M+L) cone-contrast direction for achromatic stimuli. We asked how differences in gamma among chromatic stimuli relate quantitatively to DKL cone-contrasts.

We addressed this question as follows. For the stimuli shown in Figure 4A, we calculated the cone contrasts using the principal axes of the DKL space: luminance (L+M), L-M (red-green opponency) and S-(M+L) (blue-yellow opponency). For each of those contrasts, we calculated the magnitude of the DKL cone-contrast strength irrespective of direction, for all stimuli. Among the set of chromatic stimuli, these absolute cone-contrast values were not predictive of cortical gamma, for none of the three axes (Figure 4—figure supplement 1C) (all p>0.05 uncorrected; we report the uncorrected p-values to be conservative, because the corrected p-values would be further away from the significance threshold). Note that the regression slopes show, if anything, a negative relation between absolute cone-contrast values and cortical gamma. We also performed multiple regression models that included first all three DKL axes, and in case of non-significant axes, removed these stepwise. The multiple regression analyses were in agreement with the simple regressions shown in Figure 4—figure supplement 1C.

This dataset primarily used stimuli of maximal brightness. We note that with very low luminance and therefore very low cone contrast (Figure 4B), gamma responses can decrease. This appears to be a nonlinear process (see also the sudden drop in gamma-band amplitude for responses at long wavelengths in Figure 4—figure supplement 1A).

In addition, we found that gamma oscillations were strong for stimuli with positive L-M values as opposed to negative L-M values (Figure 4—figure supplement 4D). We performed additional experiments discussed in point C2, supporting this same point.

C2) Comparison of hues with the same DKL cone-contrasts:

We have performed an additional experiment in which we kept stimuli in DKL space at precisely controlled L-M, S-(L+M) and (L+M) values (Figure 4—figure supplement 4 of revised manuscript), subsection “Controls for luminance-contrast and cone-contrast”:

“In the previous section, colored stimuli were either presented at maximum brightness or presented at the same physical luminance. We performed additional experiments in which colored surfaces were matched in terms of DKL space coordinates (Figure 4—figure supplement 4). […] This is further supported by the finding that the magnitudes of the DKL cone-contrasts for the chromatic surface stimuli shown in Figure 4A are not significantly correlated with gamma-band power (Figure 4—figure supplement 1C).”

Next, we will discuss how gamma-band power differences between chromatic and achromatic stimuli depend on cone-contrast.

C3) Achromatic stimuli on chromatic background vs. chromatic stimuli on achromatic background:

The original dataset allowed for an analysis of gamma as a function of stimulus chromaticity with matched cone-contrasts, which we have included in the revised manuscript, subsectoin “Controls for luminance-contrast and cone-contrast”:

“In another experiment, which is part of the data shown in Figure 6, we matched cone-contrasts between chromatic and achromatic stimuli. Specifically, we compared gamma-responses to a colored surface on an achromatic full-screen background with gamma responses to a corresponding achromatic surface on a chromatic full-screen background of the same respective color (e.g. red on a gray background versus gray on a red background). […] Together, these data indicate that the difference in gamma-band response between chromatic and achromatic surfaces was not due to luminance- or DKL cone-contrast relative to the full-screen background.”

C4) Quantitative comparison of drive between chromatic and achromatic surfaces:

We considered to further elaborate these analyses by modeling responses to chromatic stimuli of different sizes and structure (center mismatch or annulus) and of achromatic stimuli. Modeling the dependence of gamma on the spatial structure of chromatic stimuli would be interesting but goes beyond the scope of this study. Modeling the dependence of gamma on both chromatic and achromatic stimuli appears currently not feasible: The relation between drive by chromatic and achromatic is difficult to estimate, as extensively discussed in the original manuscript. The output MUA firing rates are higher for achromatic surfaces (in the later part of the trial). In this sense, there is not more cortical drive for chromatic surfaces. Yet, as stated by the reviewer and as we discuss, there might still be a stronger excitatory input drive from the LGN. The key complexity is that there are different types of cells providing excitatory inputs for chromatic versus achromatic surfaces in V1. For achromatic surfaces, there are excitatory inputs due to temporal luminance changes, and due to fill-in processes. Our data suggests that another essential difference between chromatic and achromatic surfaces is stronger surround inhibition for chromatic than achromatic surfaces (Figure 1-2 of original manuscript).

D) Cortical gamma does not originate from LGN.

The comments of the reviewer might imply that cortical gamma is related to LGN output in the sense that stronger LGN activity could entail stronger gamma inside the LGN. Note that previous studies indicate that in primate V1, gamma-band oscillations are generated cortically, and specifically in superficial layers of the cortex. Furthermore, theyare not found in the awake primate LGN subsection “Mechanisms of gamma-band synchronization”:

“Previous work indicates that in primate and cat V1, gamma-band oscillations are generated cortically, specifically in the superficial layers of the cortex as well as layer 4B (Bastos et al., 2014; Buffalo et al., 2011; Herculano-Houzel et al., 1999; Livingstone, 1996; Xing et al., 2012). Furthermore, they have not been detected in the LGN of awake primates (Bastos et al., 2014). Together with the results presented in this paper, this indicates that the emergence of gamma oscillations in superficial layers depends on the integration of bottom-up inputs from the LGN and layer 4 with contextual information mediated through lateral and top-down feedback. Notably, superficial layers exhibit strong lateral connectivity and are densely innervated by top-down feedback (Barone et al., 2000; Lund et al., 1993; Markov et al., 2014).”

E) Distinction between sub-types of color-selective neurons

In response to the reviewer, we have also added further detail on the different types of color-selective neurons to clarify our interpretation. In the manuscript, we mainly focused on the distinction between single-opponent vs. double-opponent responses, because this is a critical distinction for area V1 (but not LGN). The main point that we wished to convey in the original manuscript is that for large chromatic stimuli, many LGN neurons with RFs on the surface will provide strong drive to the cortex, whereas for achromatic stimuli, the drive should derive mostly from temporal luminance changes (subsection “A quantitive model relating hue dependence of γ-band oscillations to adaptation”). Thus, for large chromatic stimuli, there are neurons with RF at the surface that carry color information (even though the firing of a subset of these neurons may be suppressed by the surround, e.g. Type II modified cells). This is in agreement with the comment of the reviewer.

As mentioned by the reviewer, there are various types of color-opponent responses that can be found in area V1 and LGN. Indeed, both Type I and Type II (and modified Type II) responses may contribute to the observed effects. The way that we described single-opponent cells in the manuscript was more reminiscent of Type II responses, with RF sub-regions. Throughout the manuscript, we removed the reference to RF-subregions when we refer to the single-opponent neurons (for the Results section of Figure 7 and the corresponding discussion), because this suggests that they are spatially segregated (which is indeed not the case for Type I responses as the reviewer points out). A subset of neurons in area V1 and LGN have Type I responses. According to Tso and Gilbert, about two-thirds of color selective V1 neurons are Type II (Ts’o and Gilbert, 1988).

We have added this information with references in the Discussion section:

“It has been shown that a subset of neurons with chromatic opponencies in LGN and V1 exhibit elevated firing as compared to baseline for large chromatic surfaces, and carry information about the surfaces hue (Ts’o and Gilbert, 1988).

These cells can be subdivided into Type I, Type II and modified Type II responses. The data shown in Figure 6-7 suggests that V1 gamma oscillations are mediated by color-opponency signals. These color-selective neurons would not be active for a large achromatic stimulus (Ts’o and Gilbert, 1988). However, a subset of V1 neurons also fire to temporal luminance changes for black and white surfaces (Xing et al., 2010), and neurons might also be activated for achromatic surfaces by fill-in processes from the surround (Zweig et al., 2015).”

Reviewer #3:[…] 1) Although the number of sessions and trials per figure/analysis is reported, the reproducibility of the results across the two animals is unclear to me. Please clarify this issue: which results were reproduced for both animals? Please indicatefor each data set – how many sessions were obtained from each animal and the number of trials, for each animal. This is particularly important because the Materials and methods section indicates that the two animals performed two different behavioral tasks.

The original manuscript reported some of the main results for both monkeys separately. Following the request of the reviewer, we have included the individual results for all of the main analyses (see Figure 1—figure supplement 3, Figure 2—figure supplement 1, Figure 3—figure supplement 1, Figure 4—figure supplement 2, Figure 7—figure supplement 1, and the text corresponding to Figure 5 of the main text). All these main results were qualitatively similar across the two animals and statistically significant in each of the two monkeys. Furthermore, we have added the session numbers and trial numbers per animal in the dataset description in the Materials and methods section.

Statistical significance tests – grand analysis: it is unclear whether the authors used the combined distributions of all single trials within a dataset. Is each single trial an independent measure for chromatic vs. achromatic conditions? Or did they use the mean (across trials) for each session and then tested the statistical significance across sessions? I am asking this because usually the reproducibility and statistical significance is tested across independent recording sessions (and not the single trials).

As alluded to by the reviewer, the critical aspect in choosing the units of analysis is their independence. In studies, in which e.g. a single electrode records an isolated neuron per session, each session provides an independent unit of analysis, and testing across sessions is the appropriate approach. However, we used chronically implanted arrays of electrodes, and we mainly analyzed the LFP power spectra. The LFPs of the different electrodes are not independent, but the different trials are. Therefore, we used the trials as units of analysis. Note that we first average over trials and then over sessions, such that each session is given the same weight. Note that in many previous studies with chronic recordings, statistics have been performed by treating trials as independent (e.g. Salazar et al., (2012); Bosman et al., 2012; Kerkoerle et al., 2014; Philips et al., 2013; Martin et al., 2018). This is valid because under the null hypothesis that the different conditions do not differ (e.g. chromatic and achromatic), the trials are exchangeable between conditions (Maris and Oostenveld, 2007).

2) A central problem in this study is that the luminance/contrast/color attributes of the different stimuli were not well equated for the different conditions, which makes the interpretation very complex. This is based on the Materials and methods section and Table 1 in the Supplementary material. For example, when using chromatic stimuli the gray background (27.5 cd/m^2^) was not equal with luminance to the chromatic stimuli (Dataset 1: the color surfaces luminance varied mostly between 2.5 to 10 cd/m^2^). Thus, when color stimuli appeared over the screen there was a change also in the luminance level (compared with background). In addition, the white and black stimuli are not symmetric relative to the background. What about the luminance and L-M cone contrast: was the luminance weber contrast of the black/white stimuli adjusted to the L-M cone weber contrast of the chromatic stimuli, all relative to background? Please show the numbers. Note that Figure 4 does not provide answer for all these questions.

We thank the reviewer for these comments, which led us to perform additional control experiments and analyses to address these concerns.

The reviewer raises several connected points: (1) In the main figures of the manuscript, chromatic stimuli were not equiluminant to the background, such that effects ascribed to chromaticity could in principle have been due to luminance; (2) White and black contrast to the background was not equal, such that any difference for white versus black could have been due to this contrast difference; (3) Luminance contrasts for black and white stimuli were not equated to L-M cone contrasts for chromatic stimuli, such that differences between those achromatic and the chromatic stimuli could have been due to differences in the respective contrasts; (4) The reviewer requests the reporting of the respective numbers.

We address these four concerns point-by-point:

1) Luminance-difference of chromatic stimuli to background:

1A): In Figure 4—figure supplement 3, we have shown in a control session that there are strong gamma responses to color stimuli also in case of an equiluminant full-screen gray background. We made the same observation for the new control experiments shown in Figure 4—figure supplement 4.

1B) We added a control session, during which we presented colored stimuli from three luminance planes. Luminance was defined here as (L+M) cone-contrast in DKL space. We took values of (L+M) = +0.25, -0.25, 0 relative to the background. From each of these luminance planes, we then sampled from four different colors. The results show that gamma is reliably induced by these colors but not by achromatic stimuli, for all three luminance steps (Figure 4—figure supplement 4).

1C) In the original manuscript, we had shown that there is no correlation between Michelson-contrast of the stimuli to the full-screen background and gamma-band amplitude (Figure 4—figure supplement 2). This analysis therefore also suggests that the luminance-contrast to the background does not drive gamma-band responses.

In summary, these data suggest that the difference between chromatic and achromatic stimuli is not driven by a difference in luminance relative to the background.

These changes are summarized in the following new Subsection “Controls for luminance contrast and cone-contrast”:

“In the analyses above, we observed a strong difference in gamma-band power between chromatic and achromatic surfaces. We performed several control analyses and experiments to investigate whether this observed difference was explained by differences in DKL cone contrast or luminance contrast between chromatic and achromatic surfaces. A linear regression of gamma peak height against absolute Michelson luminance contrast (luminance stimulus luminance baseline / (luminance stimulus + luminance baseline)) across the surface stimuli shown in Figure 4B showed no significant relationship (r=-0.44,p=0.16, F-test, Figure 4—figure supplement 2; note that the relationship, if any, was negative). In an additional control experiment, we directly matched the luminance (and thereby luminance-contrast) of the achromatic and chromatic stimuli across 5 brightness values, including the full-screen background brightness and two steps of positive and negative contrast. We found that achromatic gamma-responses were much weaker than chromatic gamma-responses regardless of overall luminance level, also under these matched conditions (Figure 4—figure supplement 3A).”

2) Difference between black and white in luminance to background

2A) In Figure 4—figure supplement 3, we showed that both negative and positive luminance steps (of the same magnitude) away from the gray background do not result in detectable gamma oscillations.

2B) In a new experiment, we have made a step in positive or negative L+M cone-contrast (in DKL space) for the achromatic stimuli (and do the same for all the chromatic stimuli). This does not result in gamma-band oscillations in either luminance-direction for achromatic stimuli (Figure 4—figure supplement 4).

2C) We did not analyze the difference in gamma between black and white stimuli in much detail, as it was much weaker than for the chromatic stimuli, and in fact we found opposite patterns for the two monkeys (see response to major comment 5 of this reviewer) (Figure 4—figure supplement 2).

To conclude: While there are some differences between black and white surface stimuli when presented at very extreme luminance-contrasts (but not consistent across monkeys), gamma is weak when these luminance-steps are small and does not appear to differ between darker and brighter stimuli when these luminance-steps are matched.

3) Luminance-contrasts for black/white vs. L-M contrasts chromatic stimuli

3A) In the original manuscript, we provided a control session that equated luminance between chromatic and achromatic stimuli, for 5 different, identical luminance steps (Figure 4—figure supplement 3 of revised manuscript). The difference in gamma-band power between chromatic and achromatic stimuli was stable across these different luminance steps. Achromatic stimuli did not give rise to detectable gamma peaks, irrespective of the luminance contrast to the gray background. The main figures used maximally black/white stimuli, with correspondingly large contrasts, and thereby likely underestimated the difference between chromatic/achromatic conditions.

3B) We have performed a new set of experiments, during which we presented colored stimuli at three luminance DKL planes, one being equal luminance to the background (Figure 4—figure supplement 4A, revised manuscript). The step in the L+M direction for achromatic stimuli is comparable in magnitude to the steps in the L-M direction within the zero-luminance-contrast (L+M=0) plane. However, we find no evidence for gamma generated by achromatic stimuli, but gamma was reliably detected for chromatic stimuli in the zero-luminance-contrast plane with a similar L-M contrast (Figure—figure supplement 4) and the corresponding text in subsection “Controls for luminance-contrast and cone-contrast”:

“In the previous section, colored stimuli were either presented at maximum brightness or presented at the same physical luminance. We performed additional experiments in which colored surfaces were matched in terms of DKL space coordinates (Figure 4—figure supplement 4). […] This is further supported by the finding that the magnitudes of the DKL cone-contrasts for the chromatic surface stimuli shown in Figure 4A are not significantly correlated with gamma-band power (Figure 4—figure supplement 1C).”

3C) The original dataset recorded allowed for an analysis of gamma as a function of stimulus chromaticity with matched cone-contrasts. This analysis is now included as Figure 6—figure supplement 3 and corresponding text in subsection “Controls for luminance-contrast and cone-contrast”:

“In another experiment, which is part of the data shown in Figure 6, we matched cone-contrasts between chromatic and achromatic stimuli. Specifically, we compared gamma-responses to a colored surface on an achromatic full-screen background with gamma responses to a corresponding achromatic surface on a chromatic full-screen background of the same respective color (e.g. red on a gray background versus gray on a red background). […] Together, these data indicate that the difference in gamma-band response between chromatic and achromatic surfaces was not due to luminance- or DKL cone-contrast relative to the full-screen background.”

4) Reporting the numbers

We revised the manuscript to provide numbers for each dataset that indicate the position of each used color in DKL space in Figure 4—figure supplement 1A, Figure 4—figure supplement 2A, Figure 4—figure supplement 4, and Figure 6—figure supplement 1.

3) The authors indicate that the MU's RF were much smaller than the surfaces they have used, however I could not find the mean or median size of the RFs (as opposed to RF eccentricity that is clearly reported) for each dataset. Please report also what was the ratio between the large surface stimulus (6 deg) and the MUA RF size, for each data set.

Following the reviewers request, we have included the numbers on median RF size in the Results section:

“Classical receptive fields (RFs, referring to classical RFs unless otherwise mentioned) of the MU activity were estimated using moving bar stimuli (see Methods; monkey H: median RF eccentricity 6.2 deg, range 5.2-7.1 deg, median RF diameter 0.48 deg, range 0.26-1.88 deg; monkey A: median eccentricity 5.4 deg, range 3.2-8.5 deg, median RF diameter 0.91 deg, range 0.46-2.3 deg). Compared to a surface stimulus of 6 deg diameter, receptive fields had a median proportional diameter of 0.08 (0.04-0.31, monkey H) or 0.15 (0.08-0.38, monkey A).”

4) Late decrease in firing rate and increase in gamma-band power: what is the relation to eye movement? The authors need to control for this. Can it be that the late decrease of firing rate towards/below baseline has to do with microsaccades/small saccades that the animals start to perform at times> 200 ms? which is typical to microsaccades/saccadic inhibition?

We thank the reviewer for this comment, which led us to perform an additional control analysis. We agree with the reviewer that eye movements and in particular microsaccades play an important role in vision. We have analyzed the data in two ways: (1) Without taking microsaccades into account; (2) Excluding data recorded within 100 ms after each microsaccade. The two analyses gave essentially the same results, both for firing rates and gamma, and yielded the same difference between achromatic and chromatic stimuli (Figure 1—figure supplement 3 of revised manuscript):

“We also removed 100 ms data epochs after each microsaccade and found that the LFP results on gamma oscillations were qualitatively unchanged (Figure 1—figure supplement 3).

Control analyses in which data epochs after microsaccades were removed indicate that the late decrease in MU firing was not due tomicrosaccades (Figure 1—figure supplement 3).”

In addition, surface stimuli of 6 degree, especially the white surface that is reported to be 127 cd/m^2^ (over a gray background of 27.5 cd/m^2^), can induce changes in pupil size. A previous paper by Haynes et al. (2004) found a change in the humans' pupil due to light variation of the surface stimuli. Can the authors account for this effect in the animals?

The control analyses discussed in response to Major comment 2 of this reviewer address the issue of luminance differences. There, we also discuss control experiments (some contained in the original manuscript, some new), in which the luminance was matched between chromatic and achromatic stimuli. This should already address the concern of the reviewer.

Nevertheless, we now provide an analysis of changes in pupil size in the control experiment (Figure 4—figure supplement 3), using the original control recording sessions for luminance contrast. This analysis shows that pupil changes are relatively small and not significantly correlated with gamma-band power, indicating that our results cannot be attributed to changes in pupil size, subsection “Controls for luminance-contrast and cone-contrast” in the main text:

“We additionally used this experiment as a control for the effect of pupil size (see Materials and methods section) on gamma-band amplitudes (Figure 4—figure supplement 3B). Note that gamma responses for achromatic stimuli were weak regardless of the degree of pupil change.”

5) Many previous studies (papers from Shapley, Alonso and others) showed that V1 neurons have higher activity for black vs. white stimuli. Is there any difference in the measured neuronal results between white and black surface? Given this answer, was it justified to combine the black and white responses?

We agree with the reviewer that black and white stimuli should not necessarily be equated. We showed the difference between black and white gamma responses in Figure 4 (both panels A and B, separate experiments). We found somewhat stronger gamma for black than for white stimuli in one animal (monkey A, see Figure 4—figure supplement 2). This indeed matches the previous observation that there are stronger neural responses for black stimuli. Note however, that the other monkey did not show this effect, if anything the opposite pattern, though responses were too weak to establish this reliably. Furthermore, the difference between chromatic and achromatic stimuli in terms of gamma applies both to black and white stimuli, as well as intermediate gray values. We have added a sentence in the Results section, stating that we find stronger gamma for black than white stimuli, but that this finding is not consistent across the two monkeys:

“In one monkey (A), we found that gamma-band activity was stronger for black than for white stimuli, consistent with previous results showing stronger firing rate responses to black than white stimuli (Xing et al., 2010; Yeh et al., 2009). However, a trend in the opposite direction was observed for monkey H (Figure 4—figure supplement 2).”

6) The neural correlates of color surfaces and in particular the increased power in the gamma-band -seems to appear late in time. Firing rates appear > 200 ms and gamma-band power is computed over 600-1.3 s intervals. If we assume that color processing occurs already with the first 200 ms, can the authors clarify, what information on color surfaces, these late modulations hold?

There seems to be a misunderstanding. Most figures show gamma power quantified by an FFT calculated over a window starting at 300 ms after stimulus onset. This does not mean that there is no gamma before that, but this is a conservative approach to minimize influences of the event-related potential or generally event-related non-stationarities in the signal. The analysis window mentioned by the reviewer (0.6-1.3 s) is not used for any analysis; we guess that they refer to Figure 2B. The analysis for this figure uses the described approach: Exclusion of 0.3 s after stimulus onset or stimulus change, and therefore pools data from 0.3-0.6 s after stimulus onset with data from 0.3-0.6 s after stimulus change (corresponding to 0.9-1.2 s after stimulus onset).

The latency of contextual firing rate effects is indeed quite short. This can be seen from Figure 2—figure supplement 1C. Here, we show rapid suppression of firing rates when after 600ms of presentation of a small stimulus, we present the large stimulus. This suppression is stronger for chromatic than achromatic stimuli and this difference becomes significant at a latency of about 100ms. We have now noted this in the text of that supplementary figure and have added tick marks to enhance visibility:

“Note rapid firing suppression after onset of the large stimulus following the 0.5 degree stimulus, with a significant difference arising already after approximately 100ms.”

Finally, Figure 5 shows that gamma starts very soon after stimulus onset, with the typical drop in gamma frequency. Inspection of individual trials also reveals that there are strong gamma bouts already soon after stimulus onset (Figure 1 and Figure 3).

[Editors' note: further revisions were requestedprior to acceptance, as described below.]Essential Revisions:A few issues remain concerning the approach to addressing the main point (drive to cortex). In reply to major comment 2 the authors (point A and C4 of response) state that large chromatic surfaces do not lead to strong V1 firing rates but do yield strong gamma. If the authors mean that the firing rates in their study seem similar for chromatic and achromatic stimuli this seems correct, however there it likely that their unit activity recordings are biased away from layer 4, and there is also is reasonably good evidence that chromatic stimuli will yield stronger overall firing rates for uniform fields than will achromatic stimuli (see for review Schluppeck and Engel, (2002). I am not sure why the authors argue that chromatically varying center-surround stimuli will produce as strong a response from LGN (or V1) standard units (point (B) in response – they will certainly be less effective for all but double opponent cells, which are not present in LGN, and are rare in V1. The same goes for chromatic edges. While the points above may seem semantic they are still important – they mean that the authors have thoroughly ruled out some variant of 'drive' as the primary correlate of gamma in these experiments.

We thank the reviewer for these comments. The reviewer makes two major points about the circuit mechanisms underlying our findings.

The first point is that there may be stronger excitatory input drive from the LGN for chromatic than achromatic stimuli for units with receptive fields at the uniform region of the surface, even though V1 firing rates in superficial layers may be lower for chromatic stimuli. We agree with this interpretation, which we had already put forward in the Discussion of the original and revised manuscript (page 9). In the newly revised manuscript have further improved the clarity with which this is stated:

“We asked whether there are differences in V1 gamma-band synchronization and firing responses between chromatic and achromatic stimuli. A previous voltage-sensitive dye imaging study has shown differences in V1 responses to chromatic compared to achromatic surfaces (Zweig et al., 2015). […] Because the fMRI signal correlates not only with spiking activity, but also with gamma responses (Logothetis and Wandell, 2004,, Ekstrom, 2010, Maier et al. 2008, Thomsen et al., 2004, Visnawathan and Freeman, 2007, Nir et al., 2007, Scheeringa et al., 2016, for visual gamma in particular Bartolo et al., 2011, Niessing et al., 2005), it remains unclear how these fMRI findings are related to the present findings.”

The second point pertains to the comparison between uniform stimuli and stimuli with a center-surround mismatch. In the original and revised manuscript we have shown that V1 firing rates are much higher for the case of a center-surround mismatch (annulus or blob) or at the edge of the uniform stimuli, as compared to the uniform region of homogeneous surface stimuli. Note that this rate difference is present from the onset of the responses. At the same time, gamma oscillations were dramatically reduced for those stimuli that gave rise to high V1 firing rates. These firing rates were, in contrast to uniform stimuli, well above baseline firing rates. We have added further discussion on how this finding relates to previous results on chromatic opponencies in V1 in the Discussion section of the newly revised manuscript:

“Overall, our data suggests that both sufficient drive and spatial predictability are the necessary ingredients for the generation of V1 gamma. There are several cases in our manuscript where differences in V1 gamma are dissociated from differences in firing rate. […] As opposed to these cases where high firing rates are accompanied by weak gamma, we also present cases where stimuli have high spatial predictability, but where gamma is weak or absent likely due to low drive (Figure 1 achromatic stimuli, Figure 6 chromatic stimuli after prolonged adaptation).”